# Demystifying GNN Distillation by Replacing the GNN

## Abstract

It has recently emerged that Multilayer Perceptrons (MLPs) can achieve excellent performance on graph node classification, but only if they distill a previously-trained Graph Neural Network (GNN). This finding is confusing; if MLPs are expressive enough to perform node classification, what is the role of the GNNs? This paper aims to answer this question. Rather than suggesting a new technique, we aim to demystify GNN distillation methods. Through our analysis, we identify the key properties of GNNs that enable them to serve as effective regularizers, thereby overcoming limited training data. We validate our analysis by demonstrating an MLP training process that successfully leverages GNN-like properties without actually training a GNN.

## 1 Introduction

Node classification tasks naturally occur when we wish to classify graph-structured data, such as paper citation networks (Sen et al., 2008; Namata et al., 2012) or product co-purchase networks (Shchur et al., 2018). Most state-of-the-art node classification methods use Graph Neural Networks (GNN) (Kipf & Welling, 2016; Hamilton et al., 2017; Velickovic et al., 2017). GNNs exploit the context of the neighborhood of each node to determine its class. Their architecture uses message passing to transfer information across many nodes. As GNNs consider large neighborhoods, their training and inference times are considerably higher than methods that consider only the node features.

Therefore, several research efforts have attempted to find simpler alternatives to GNNs that do not require message passing. Distillation methods, starting with the seminal work by Zhang et al. (2021), propose to replace GNNs at test time with simple node-level MLPs. These methods first train a GNN on the training data, then use the labels predicted by the GNN as distillation targets for training a node-level MLP. At test time, these methods only use the faster node-level MLP. Remarkably, distillation methods sometimes outperform GNNs, despite MLPs not directly using the graph structure.

This study investigates the role of GNNs in the distillation process and finds that their primary contribution lies in enforcing regularization that preserves homophily, the tendency for adjacent nodes to share labels. Our analysis reveals that GNNs provide an effective implicit regularization which assists in reducing overfitting, especially in datasets characterized by significant homophily and very limited training data.

Some distillation methods also propose incorporating structure information into the input features of the MLP student model using learnable positional embeddings. While effective, it is unclear which structural aspects of the graph these features encode. We present an explainable alternative descriptor that encodes the neighborhood characteristics of each node.

To validate that the GNN's implicit regularization is key to the success of distillation methods, we compare distillation methods to an alternative approach of training MLPs with direct regularization. This regularization strategy consists of three key components: (i) a loss encouraging smoothness between the label predictions of neighboring nodes, (ii) iterative pseudo-labeling of the observed unlabeled nodes, and (iii) a neighborhood label histogram descriptor for encoding local context. Notably, this approach does not require training or evaluating GNNs and demonstrates a high correlation with the performance of distillation methods on commonly used datasets, such as citation and co-purchase networks.

Our key contributions are:

- Through theoretical and empirical study, we demonstrate that the role of GNNs in distillation methods is to act as regularizers rather than to increase expressivity.

- Demystifying GNN regularization by replacing it with explicit regularization terms.

- Suggesting label histogram as a more explainable alternative to positional embedding.

## 2 RELATED WORKS

**Graph Neural Networks (GNNs)** have emerged as a prominent tool in the domain of graph machine learning (Bruna et al., 2013; Defferrard et al., 2016; Li et al., 2019; Chen et al., 2020b). These neural networks use aggregations of features from the local context of each node at successive layers. For example, Graph Convolutional Networks (*GCN*) (Kipf & Welling, 2016) extend traditional convolution operations from the Euclidean domain to operations on graphs. *GraphSAGE* Hamilton et al. (2017) uses arbitrary aggregation functions while also concatenating the features prior to the aggregation. GAT (Velickovic et al., 2017), GTN (Yun et al., 2019), and HAN (Wang et al., 2019b) generalize attention layers and transformers to graphs. Many GNNs are formulated into a unified framework called *Message Passing* Neural Networks (Gilmer et al., 2017).

**Knowledge distillation of GNNs.** Addressing challenges related to memory consumption and latency, several methods have been proposed to distill knowledge from a large pre-trained GNN teacher model to a smaller student model. The student model can be either a smaller GNN model (Lee & Song, 2019; Yang et al., 2020; Yan et al., 2020; Tian et al., 2023; Guo et al., 2023), or a structure-agnostic model (Wu et al., 2023a;b). One such method, *GLNN* (Zhang et al., 2021), trains an MLP model to predict soft-labels obtained from a pre-trained GNN. Another approach, *NOSMOG* (Tian et al., 2022), uses the same underlying method with the addition of adversarial feature augmentation loss and *Similarity Distillation* of hidden features. NOSMOG also utilizes the graph structure by concatenating positional features obtained using DeepWalk (Perozzi et al., 2014). While NOSMOG offers better accuracy results than standard GLNN, it suffers from higher latency induced by positional feature computation. PGKD (Wu et al., 2023c) distills graph structural information from GNNs to MLPs via prototypes in an edge-free setting. Orthi-Reg (Zhang et al., 2023) mitigates the dimensional collapse of MLPs by explicitly encouraging orthogonal node representations during training. *CPF* (Yang et al., 2021) also uses a non-GNN student model, although the student still relies on iterative label propagation during inference, which increases the inference running time.

**Node classification without GNN.** Various techniques beyond Graph Neural Networks have been developed. Among them is *Graph-MLP* (Hu et al., 2021), which trains an MLP model with a neighbor contrastive loss. While this approach bears some resemblance to the consistency loss employed in our work, our consistency loss encourages similarity among neighboring nodes and is not contrastive. Another method, *Correct and Smooth* (C&S) (Huang et al., 2020), leverages the correlation between neighbors' labels to enhance a shallow MLP predictor. Unlike our study, which focuses on the training process of MLPs, it refines predictions post-training through label propagation. The applicability of the C&S method to the inductive case (where new nodes are added to the graph during test time) is limited, and it focuses on supervised rather than semi-supervised settings. Also, in contrast to the heavy reliance on labels by C&S, a significant aspect of our study examines how GNNs address the challenges arising from a small training set in semi-supervised scenarios.

**Semi-supervised learning.** SSL is an approach for leveraging unlabeled data, often used in scenarios where the size of the training set is small. A popular SSL method, termed *pseudo-labeling*, uses the model's predictions as labels for training (McLachlan, 1975; Rosenberg et al., 2005; Lee et al., 2013; Xie et al., 2020). Another prominent SSL approach is consistency regularization (Bachman et al., 2014; Sajjadi et al., 2016; Laine & Aila, 2016), where the model is enforced to maintain consistent predictions through random augmentation of its input. FixMatch (Sohn et al., 2020) combines these ideas in a simple manner.

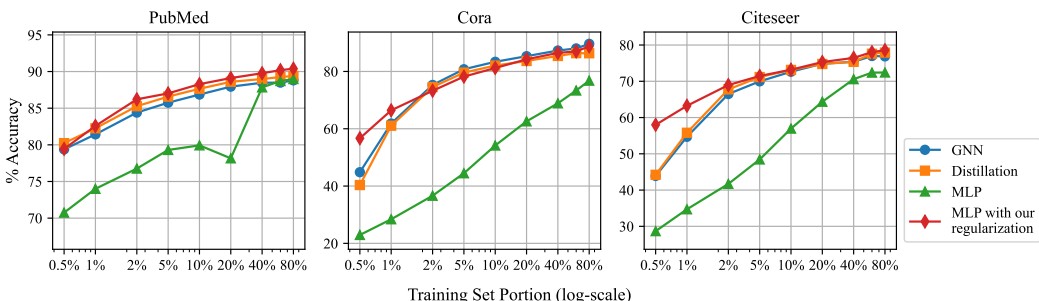

Figure 1: The performance gap between vanilla MLP model (green) to GraphSAGE GNN (blue) diminishes as the training set size increases. Our regularization (red) refers to an MLP model that is trained with consistency loss and smoothed pseudo labels. Distillation (orange) refers to an MLP trained with GNN distillation loss. Our regularization approach emulates the GNN distillation.

## 3  WHY ARE DISTILLATION METHODS SUCCESSFUL?

Distillation methods have recently challenged the existing paradigm in node classification. The standard practice with GNNs is to train the model on all the labeled nodes in the graph and use the same model for node classification at test time. Distillation methods remove the need for using a GNN at test time, although they still require training a GNN for teaching the student model using unsupervised nodes. They train an MLP to predict the labels of each node based on node features only, without considering the features of its neighbors. Remarkably, distillation methods are competitive with GNNs on popular citation and co-purchase network benchmarks. This result is confusing, as the graph structure appears beneficial during training but not during test time. This begs the question: *Why are distillation methods so successful?*

To address this question, we investigate whether the power of the GNN comes from the increased expressivity of message passing or a useful inductive bias. As a quick test, we plot the node classification accuracy of both GNNs (specifically GraphSAGE) and node-level MLPs as a function of the training set size (Fig. 1). The observed trend indicates that with an increase in training size, the performance gap between node-level MLPs and GNNs diminishes. This suggests that MLPs overfit due to small training sizes on popular node classification datasets, while GNNs have better regularization (i.e., they have a useful inductive bias).

Next, we examine the gap between GNNs and MLPs that are trained through distillation of GNNs. Here, we observe that the gap between the two models is narrow, even for small labeled training sets. These experimental findings lead us to conclude that: *the challenge in the examined dataset lies not in increasing model expressivity, but rather in decreasing model sample complexity.* GNNs overcome this challenge through a useful inductive bias, while distillation overcomes it by increasing the size of the training set with GNN pseudo-labels.

We hypothesize that GNNs improve the optimization of MLPs due to two key properties:

- GNNs act as consistency regularizers due to their tendency to produce smoothed predictions along edges.

- GNNs benefit from unlabeled data through the message passing mechanism.

In the following sections, we will empirically test this hypothesis. We will analyze the benefits of these properties and examine how we can emulate their effects without the need to train a GNN.

Table 1: Notation summary

| Notation | Explanation | Notation | Explanation |
|----------|-------------|----------|-------------|
| $\mathcal{G}$ | Graph | $\mathcal{N}(v)$ | Set of immediate neighbors of $v$ |
| $\mathcal{V}$ | Set of nodes | $\mathcal{N}^\ell(v)$ | Set of nodes $u \in \mathcal{V}$ s.t. $d(v,u) \leq \ell$ |
| $\mathcal{V}_{train}$ | Training set, subset of $\mathcal{V}$ | $X$ | Feature matrix in $\mathbb{R}^{n \times d}$ |
| $\mathcal{V}_{val}$ | Validation set, subset of $\mathcal{V}$ | $\mathbf{x}_i$ | The $i$-th row of $X$ |
| $A$ | Adjacency matrix | $\mathbf{y}_i$ | One-hot label of $v_i$ in $\{0,1\}^C$ |
| $C$ | Number of classes | $\Psi$ | Student MLP |
| $d(v,u)$ | Length of shortest path from $v$ to $u$ | $\Phi$ | Teacher GNN |

## 4 REPLACING GNN DISTILLATION WITH EXPLICIT REGULARIZATION

### 4.1 PRELIMINARIES

**Notation.** We are given a graph $\mathcal{G} = (\mathcal{V}, A)$ where $\mathcal{V}$ is a set of nodes $\{v_1, ..., v_n\}$ and $A$ is the adjacency matrix, i.e.,

$$A_{ij} = \begin{cases} 1 & v_i \text{ is directly connected to } v_j \\ 0 & \text{else} \end{cases}$$

We ignore self-loops in the graph, hence $A_{ii} = 0$ for all $i \in \{1, ..., n\}$. In addition, we are given a node feature matrix $X \in \mathbb{R}^{n \times d}$ where its $i$-th row is the feature vector of node $v_i$ and is denoted by $\mathbf{x}_i$. We define the training set $\mathcal{V}_{train} \subset \mathcal{V}$ and the validation set $\mathcal{V}_{val} \subset \mathcal{V}$. For each $v_i \in \mathcal{V}_{train} \cup \mathcal{V}_{val}$, we are given a label $\mathbf{y}_i \in \{0,1\}^C$ encoded as a one-hot vector, where $C$ is the number of classes. We denote by $d(u,v)$ the length of the shortest path in $\mathcal{G}$ between the nodes $u$ and $v$. Furthermore, we denote the set of nodes that can be reached from $v$ with paths of distance no longer than $\ell$ by $\mathcal{N}^\ell(v)$, i.e, $\mathcal{N}^\ell(v) = \{u \in \mathcal{V} | d(v,u) \leq \ell\}$. We omit the superscript for $\ell = 1$, denoting $\mathcal{N}^1(v)$ as $\mathcal{N}(v)$.

**Task.** Our goal is to predict the labels of all the nodes in $\mathcal{V}/(\mathcal{V}_{train} \cup \mathcal{V}_{val})$. Following common practices, we only use $\mathcal{V}_{train}$ for optimizing the model weights and $\mathcal{V}_{val}$ for hyper-parameter selection. Note that we described the transductive settings, where all the nodes of the test set are accessible during training. We describe the inductive setting, where some nodes of the test set are not present during training, along with corresponding experiments in Sec. A.7.1.

### 4.2 THEORETICAL BACKGROUND: GNN DISTILLATION

Distillation methods, such as GLNN (Zhang et al., 2021), take a two-step training approach. First, they train a teacher GNN model, $\Phi$, on the training set with the standard cross-entropy loss between the prediction of the model and the ground truth (GT) labels. Formally, the loss is given by:

$$\mathcal{L}_{GT}(\Phi) = \sum_{v_i \in \mathcal{V}_{train}} CE(\mathbf{y}_i, \Phi(v_i)) \tag{1}$$

Where the Cross Entropy (CE) between two distribution vectors $p$ (observations) and $q$ (model's predictions) is $CE(p,q) = -\sum_{c=1}^{C} p(c) \log(q(c))$.

After obtaining a fully trained GNN model, they freeze its weights and proceed to train a student MLP model, $\Psi$. At the core of distillation methods is training $\Psi$ with the following loss $\mathcal{L}(\Psi)$ [1]:

$$\mathcal{L}_{distill}(\Psi) = \sum_{v_i \in \mathcal{V}} CE(\Phi(v_i), \Psi(v_i)) \tag{2}$$

$$\mathcal{L}(\Psi) = \mathcal{L}_{GT}(\Psi) + \gamma \cdot \mathcal{L}_{distill}(\Psi) \tag{3}$$

---

[1]In some literature, the KL-divergence function is used in $\mathcal{L}_{distill}$. However, we use cross-entropy (CE) for convenience. Minimizing CE is equivalent to minimizing KL, since the weights of $\Phi$ are fixed during training and we have $KL(\Phi(v_i), \Psi(\mathbf{x}_i)) = CE(\Phi(v_i), \Psi(\mathbf{x}_i)) - H(\Phi(v_i))$ where $H(p)$ is the entropy of $p$.

Table 2: Optimizing MLPs with consistency loss improves accuracy by 13%, on average. Using also iterative training with smoothed pseudo-labeling highly correlates with the performance of GNN knowledge distillation (GLNN).

| Dataset | GLNN | MLP | MLP + Consistency | MLP + Consistency + Iterations |
|---------|------|-----|-------------------|-------------------------------|
| Cora | $80.54 \pm 1.35$ | $56.88 \pm 4.87$ | $79.21 \pm 4.68$ | $79.76 \pm 1.48$ |
| Citeseer | $71.77 \pm 2.01$ | $53.59 \pm 4.46$ | $71.12 \pm 2.20$ | $73.30 \pm 2.09$ |
| Pubmed | $75.42 \pm 2.31$ | $64.03 \pm 2.32$ | $71.24 \pm 1.71$ | $75.15 \pm 3.56$ |
| A-computer | $83.03 \pm 1.87$ | $67.32 \pm 3.07$ | $75.29 \pm 1.88$ | $80.04 \pm 3.63$ |
| A-photo | $92.11 \pm 1.08$ | $77.75 \pm 2.45$ | $89.06 \pm 1.20$ | $92.03 \pm 1.79$ |
| Mean | $80.57 \pm 1.78$ | $63.91 \pm 3.59$ | $77.18 \pm 2.60$ | $80.06 \pm 2.67$ |

where $\gamma$ is a hyper-parameter.

We showed in Sec. 3 that training a simple node-level classifier tends to overfit on standard node-classification datasets due to very limited training set sizes. However, training these models with the distillation loss reduces overfitting significantly.

**Connection between GCN and Laplacian smoothing.** Prior work (Li et al., 2018; Chen et al., 2020a) has shown that GNNs produce predictions with a high positive correlation between adjacent nodes. For example, a GCN model shares a connection with Laplacian smoothing. In each layer of a GCN model, the feature matrix $X$ is transformed into $Z$ by averaging the features of each node with its neighbors, i.e., $Z = \hat{A}X$, where $\hat{A} = \widetilde{D}^{-\frac{1}{2}}\widetilde{A}\widetilde{D}^{-\frac{1}{2}}$ is the symmetrically normalized adjacency matrix, $\widetilde{A}$ is the adjacency matrix with self-loops, and $\widetilde{D} = \sum_j \tilde{A}_{ij}$ is the degree matrix. After obtaining $Z$, the GCN model applies a linear layer.

The matrix form of the Laplacian smoothing operation (Taubin, 1995) over $X$ is $\left((1 - \gamma)I + \gamma\widetilde{D}^{-1}\tilde{A}\right)X$. By setting $\gamma = 1$ and using the symmetrically normalized Laplacian, $\widetilde{D}^{-\frac{1}{2}}\widetilde{A}\widetilde{D}^{-\frac{1}{2}}$, instead of the normalized Laplacian $\widetilde{D}^{-1}\tilde{A}$, we get $\widetilde{D}^{-\frac{1}{2}}\tilde{A}\widetilde{D}^{-\frac{1}{2}}X$. This modified Laplacian operation equals the transformed inputs $Z$ in the GCN. Li et al. (2018) also demonstrated that repeatedly applying Laplacian smoothing results in uniform feature vectors for each connected component. Thus, GCNs are geared toward homophilic graphs where smoothing is beneficial.

To examine our hypothesis that the smooth predictions of the GNN act as regularization that reduces overfitting, we take the first step in replacing the GNN in the distillation with explicit components. We replace the term $\mathcal{L}_{distill}$ with a direct regularization term that we call the *consistency loss*.

### 4.3 CONSTRUCTING EXPLICIT REGULARIZATION

#### 4.3.1 CONSISTENCY LOSS

To study the distillation term, we propose incorporating an explicit regularization term instead of the implicit one provided by the GNN. Concretely, we incorporate a homophilic prior on the node predictions using a consistency loss. In many node classification tasks, such as predicting attributes of academic papers in a citation network or attributes of products in a co-purchase network, neighboring nodes typically have the same label. Homophilic priors enforce label consistency between neighboring nodes. In practice, our model does not output a single label but rather a probability distribution over the classes for each node. Therefore, to enforce consistency, we use a probability discrepancy measure between the predictions of adjacent nodes. Specifically, we compute the average cross-entropy between the predicted label distribution for a node and each of its neighbors. This loss term encourages the model to produce consistent predictions for adjacent nodes.

We formulate the consistency loss term as:

$$\mathcal{L}_{consist}(\Psi) = \sum_{v_i \in \mathcal{V}} \left( \frac{1}{|\mathcal{N}(v_i)|} \sum_{v_j \in \mathcal{N}(v_i)} CE\left(\Psi\left(\mathbf{x}_j\right), \Psi\left(\mathbf{x}_i\right)\right) \right) \tag{4}$$

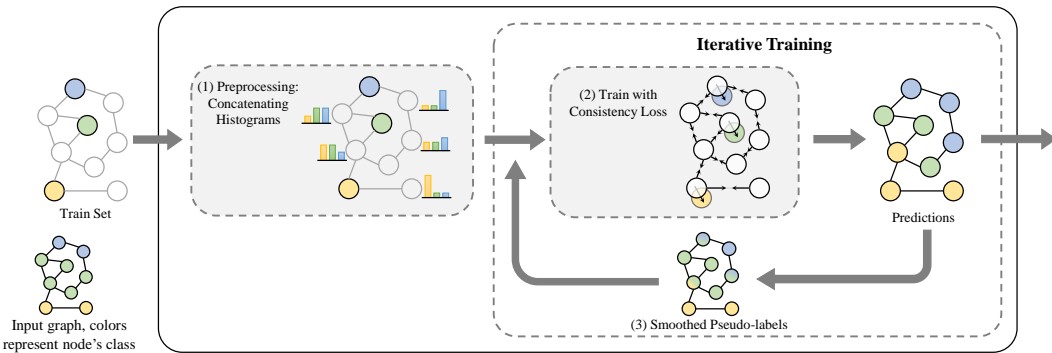

Figure 2: Our analysis consists of three elements: (1) augmenting the node features by concatenating them with the histogram of nearby node labels (Sec. 5), (2) training with consistency loss in addition to the standard cross-entropy classification loss (Sec 4.3.1), and (3) iterative training with smoothed pseudo-labels (Sec. 4.3.2).

The complete loss function is:

$$\mathcal{L}\left(\Psi\right) = \mathcal{L}_{GT}\left(\Psi\right) + \gamma \cdot \mathcal{L}_{consist}\left(\Psi\right) \tag{5}$$

where $\gamma$ is a hyper-parameter controlling the consistency regularization strength.

**Theoretical connection to distillation.**   We present an extension of the theory discussed in Sec. 4.2. The following proposition draws a connection between consistency loss and distillation methods. A formal proof is provided in the Appendix in Sec. A.2.

**Proposition 1.** *Let $\Psi$ be a linear classifier with weights matrix $W$. Define a corresponding GNN model, $\Phi$, as a 1-layer GCN model[2] that uses the same weights $W$. Then the linear model $\Psi$ that minimizes the loss $\mathcal{L}_{consist}$ is equivalent to the model that minimizes the GNN distillation loss, where the GNN model $\Phi$ acts as the teacher and the linear model $\Psi$ acts as the student.*

**Datasets.**   We use a selection of datasets commonly utilized in the graph learning community: Cora, CiteSeer, PubMed, A-Computers, and A-Photo (Sen et al., 2008; Namata et al., 2012; Shchur et al., 2018). Additional details and statistics are provided in the Appendix. Following previous works (Yang et al., 2021; Zhang et al., 2021; Tian et al., 2022), we consider only the largest connected component of each graph dataset and treat the edges as undirected. The datasets are partitioned by randomly sampling 20 instances per class for training, 30 instances per class for validation, and using the remaining nodes as the test set.

**Evaluation.**   We run each experiment 10 times and report the accuracy on the test set as well as its standard deviation. As seen in Tab. 2, incorporating the consistency loss into the optimization process of the MLP improves its accuracy by $13\%$, on average. However, it still underperforms an MLP model that was trained with GNN distillation (GLNN). Therefore, we study which additional GNN components might be needed to provide a simple MLP with the inductive bias of the GNN.

### 4.3.2 PSEUDO-LABELING ITERATIONS

As demonstrated by many works in the semi-supervised regime (Rosenberg et al., 2005; Sohn et al., 2020), using unlabeled samples is crucial when we have very limited training data. Utilizing unlabeled nodes is particularly beneficial for distillation methods, as most standard node classification benchmarks are effectively semi-supervised. I.e., their training set is very small (sometimes as small as 0.3% of the total number of nodes).

Distillation methods benefit from unlabeled nodes in two ways. Firstly, during the training of the teacher GNN, the message-passing mechanism inherently uses nodes that are not necessarily in the training set. Secondly, the entire graph is passed through the GNN to obtain soft pseudo-labels for all the nodes. Consequently, all the nodes of the graph are involved in the training of the student.

---

[2]This procedure is discussed by Yang et al. (2022)

To validate that unlabeled nodes benefit MLP model training on standard benchmarks, we incorporate ideas inspired by semi-supervised research. To expand the effective training set, we add predicted labels for some unlabeled nodes. Specifically, we propose using iterative training.

Before each iteration, we add to the training set the nodes on which the model made high-confidence predictions along with the ground truth training nodes. We denote the class predictions on unlabeled nodes on which the model was confident as the *pseudo-labels*. This set is used for training the model in the following iteration. Ground-truth training nodes remain in every iteration, but high-confidence pseudo-labeled nodes are recomputed. If a high-confidence node becomes a low-confidence node in a later iteration, we will exclude it from the training set unless its ground-truth label was provided. This adaptive mechanism allows the model to correct early errors as training progresses. The training set for iteration $I + 1$ is defined by the following rule:

$$\mathcal{V}_{train}^{I+1} = \mathcal{V}_{train} \cup \{v_i | \max_{j=1,...,C} \left( \Psi^I \left( \mathbf{x}_i \right) \right)_j > \tau \} \tag{6}$$

Where $\Psi^I$ is the model that was trained on the training set $\mathcal{V}_{train}^I$, and $\tau$ is the confidence threshold.

**Pseudo-label smoothing.** As discussed in Sec. 4.2, GNNs tend to produce similar predictions to neighboring nodes. Accordingly, we observed that on many popular datasets, manually smoothing the model predictions on the target node with the predictions on the neighboring nodes improves its performance. As GNN distillation methods aim to avoid using the graph structure during inference, we also want to avoid using it. Therefore, we perform prediction smoothing (that relies on the graph structure) to the pseudo-labels only during the iterative training. We found that with this strategy, the model achieves similar final accuracy to that achieved through test-time smoothing, without actually smoothing at test time. This observation suggests that the model learns to integrate the homophilic prior into its predictions.

The smoothing technique applied to pseudo-labels involves generating predictions $\hat{Y} \in \mathbb{R}^{n \times C}$ for all nodes after each training iteration. Each row of $\hat{Y}$ represents the predicted distribution vector for a specific node. Subsequently, an adjusted prediction $Y^*$ is computed for each node by taking a weighted average between its own prediction and the average prediction of its neighbors. The weighting factor $\lambda$, determined empirically using the validation set, is introduced in the smoothing process through the equation:

$$Y^* = \lambda \cdot \hat{Y} + (1 - \lambda) \cdot \hat{A}\hat{Y} \tag{7}$$

Here, $\hat{A}$ denotes the normalized adjacency matrix, such that the $i$-th row of $\hat{A}\hat{Y}$ represents the average prediction of the neighbors of node $i$. The resulting $Y^*$ serves as the refined prediction used in the pseudo-labeling strategy described above.

As seen in Tab. 2, the proposed strategy for explicitly using the unlabeled nodes matches the performance of an MLP that was trained with GNN distillation.

# 5 EXPLICIT GRAPH STRUCTURE FEATURES FOR MLP CLASSIFIERS

Some distillation methods have proposed incorporating structural information into the input features of the MLP student model. For example, NOSMOG (Tian et al., 2022) incorporates a positional embedding, DeepWalk (Perozzi et al., 2014), in its features. This embedding allows the classifier to learn a connection between a node's position in the graph and the labels of the nodes around it. While effective, it is unclear which structural aspects of the graph these features encode. In this section, we propose an explicit approach for incorporating structural information.

## 5.1 HISTOGRAMS OF NEIGHBORING LABELS

The neighborhood of a node may help infer further information about its label. To utilize this, we propose concatenating a descriptor of neighboring labels to the input of the predictor, thereby incorporating positional data. Specifically, for each node $v$, the descriptor is a weighted histogram of the labels of all nodes with a path to $v$ of length at most $\ell$. The weight assigned to each node in the histogram is determined by the minimal path length to $v$.

Table 3: Our label-histograms closely match distillation methods, while utilizing the structural features in a more explainable way. Results show accuracy (higher is better).

| Dataset | SAGE | GLNN | NOSMOG | Ours |
|---|---|---|---|---|
| Cora | $80.52 \pm 1.77$ | $80.54 \pm 1.35$ | $83.04 \pm 1.26$ | $82.92 \pm 1.15$ |
| Citeseer | $70.33 \pm 1.97$ | $71.77 \pm 2.01$ | $73.78 \pm 1.54$ | $75.64 \pm 1.68$ |
| Pubmed | $75.39 \pm 2.09$ | $75.42 \pm 2.31$ | $77.34 \pm 2.36$ | $77.22 \pm 2.49$ |
| A-computer | $82.97 \pm 2.16$ | $83.03 \pm 1.87$ | $84.04 \pm 1.01$ | $81.03 \pm 1.60$ |
| A-photo | $90.90 \pm 0.84$ | $92.11 \pm 1.08$ | $93.36 \pm 0.69$ | $93.06 \pm 1.56$ |
| Arxiv | $70.92 \pm 0.17$ | $72.15 \pm 0.27$ | $71.65 \pm 0.29$ | $71.35 \pm 0.25$ |
| Products | $78.61 \pm 0.49$ | $77.65 \pm 0.48$ | $78.45 \pm 0.38$ | $81.71 \pm 0.26$ |
| Mean | $78.52 \pm 1.56$ | $78.95 \pm 1.52$ | $80.24 \pm 1.27$ | $80.42 \pm 1.49$ |

Unlike GNN message passing, which requires computing hidden features for the entire neighborhood, this descriptor only requires simple counting of the neighborhood labels. This is a much weaker requirement than running a GNN over the entire neighborhood.

The histogram descriptor $\mathbf{h}_i$ for a node $v_i$ is calculated by first computing $\mathbf{h}'_i$ as follows:

$$\mathbf{h}'_i = \sum_{v_j \in \mathcal{N}^\ell(v_i) \cap \mathcal{V}_{train}} \left( \alpha^{d(v_i, v_j)} \cdot \mathbf{y}_j \right) \tag{8}$$

Here, $\alpha \in [0, 1]$ is a hyper-parameter controlling the relative importance of far away nodes. Since $\mathbf{y}_j$ is a one-hot vector in $\{0, 1\}^C$, $\mathbf{h}'_i$ represents a weighted sum of labels from nodes within a local context of $v_i$, with the size of the context determined by $\ell$.

Subsequently, to obtain a normalized histogram, $\mathbf{h}_i$, we divide $\mathbf{h}'_i$ by its sum. This descriptor is concatenated to the original input vector $\mathbf{x}_i$.

$$\mathbf{h}_i = \frac{\mathbf{h}'_i}{\sum_{j=1}^C \mathbf{h}'_{ij}} \tag{9}$$

**Approximating label histograms in large graphs.** The requirement of determining the distance between each node in the training set and all other nodes in the graph is a task with a computational complexity of $\mathcal{O}(|\mathcal{V}_{train}| \cdot |E|)$, where $|E|$ is the number of edges in the graph (in the case of more edges than nodes). In the standard setting for node-level classification tasks, the size of $\mathcal{V}_{train}$ is often very small, so computing the histograms is feasible. We include in our evaluation two larger graphs from the Open Graph Benchmark (Hu et al., 2020): *ogbn-arxiv* (169K nodes) and *ogbn-products* (2.4M nodes). As these datasets have larger training sets, we use a modified approximation of Eq. 8 for them. This approximation is much faster to compute using graph convolution operations. Further implementation details and ablations are provided in the Appendix in Sec. A.3.

## 5.2 Evaluation

We evaluate the effectiveness of label histograms combined with our approach for optimizing MLPs, i.e., with consistency loss and iterative training. As seen in Tab. 3, using the label-histogram achieves results similar to distillation methods that leverage learnable positional features. We compare it to two state-of-the-art methods: (1) GLNN, a standard distillation method, and (2) NOSMOG, which adds an adversarial feature augmentation loss, similarity distillation of hidden features, and fuses positional encoding to the input. We also compare to the teacher used in KD methods - GraphSAGE with a GCN aggregation strategy. As in previous experiments, we use common benchmarks, running each experiment 10 times and reporting the accuracy and its standard deviation. Additionally, we assess the model in an inductive setting, where specific nodes in the test set are not included during the training phase. The results of this evaluation are provided in Section A.7.1

## 6 Discussion

**Mechanistic explanation.** Our analysis examines the role of GNNs in training MLPs with knowledge distillation by replacing implicit components with explicit ones. This provides a way to obtain

a mechanistic understanding of a complex, black-box approach. Specifically, we trained the MLP with 3 components inspired by GNNs, and the results indicate that our explicit components behave similarly to distillation methods that use GNNs. While other factors may contribute to the effectiveness of distillation methods, the high correlation between their results and those of our explicit approach suggests that we uncover some of their key properties. Furthermore, the components we identified are beneficial on their own.

Table 4: Leveraging the homophily prior, through consistency loss and pseudo-label smoothing, improves the model's accuracy by 2.4% on average.

| Dataset | W.o. Homophily Prior | $\Delta$ | Edge Homophily | Node Homophily |
|---|---|---|---|---|
| cora | 79.64 | -3.28 | 84.3% | 86.2% |
| citeseer | 73.97 | -1.67 | 79.5% | 80.3% |
| pubmed | 76.61 | -0.61 | 83.8% | 86.5% |
| a-computer | 79.21 | -1.82 | 78.3% | 81.7% |
| a-photo | 87.44 | -5.62 | 83.2% | 86.1% |
| Arxiv | 71.35 | 0 | 67.8% | 70.7% |
| Products | 78.07 | -3.64 | 80.8% | 81.7% |
| Mean | **78.04** | **-2.38** | | |

**Homophily prior.**   The use of the homophily prior, which posits that neighboring nodes has positively correlated labels, is reflected in our analysis in two ways: (1) The inclusion of a consistency loss, which encourages the model to maintain correlation among neighboring nodes, and (2) smoothing the pseudo-labels used during iterative training, further incentivizing the model to provide smoothed predictions across edges. Here, we study the advantages of incorporating this prior.

As depicted in Tab. 4, incorporating these two elements resulted in an average improvement of 2.4%. This can be explained by the homophilic tendencies of the dataset sources. For instance, in citation networks, it is reasonable to expect that papers in the same field may cite each other. Similarly, in co-purchasing networks, it is plausible that customers tend to buy items from the same category at the same time. The *ogbn-arxiv* dataset is an outlier as it is not positively affected by the homophily prior. Notably, this dataset has the largest proportion (53.7%) of labeled training nodes. As the number of labels is sufficient, the advantage of using additional priors such as homophily decreases.

**Computational efficiency.**   While our primary objective is to understand the mechanisms of distillation methods, our analysis also provides strategies to optimize MLPs for more efficient node classification in graphs, eliminating the need for GNNs. This approach has the potential to inspire future research focused on developing more efficient methods. Furthermore, the label histograms utilized in our analysis to evaluate the benefits of local context can be computed on a CPU and do not necessitate the learning of embeddings, in contrast to our baseline approach.

**Influence of node degree.**   The label histogram and consistency loss in our approach rely on neighborhood data. To assess the impact of node degree on model performance, we evaluated nodes with varying degrees, as illustrated in App. Fig 5. The results indicate that our optimization process for MLPs closely aligns with the patterns observed in GNN distillation. In the *a-computer* dataset, the accuracy appears to exhibit a stronger positive correlation with node degree compared to other datasets. Yet, generally there is no strong connection between the performance and the node degree across datasets.

## 7   LIMITATIONS

**Heterophilic graphs.** Current methods of GNN knowledge distillation have demonstrated success mostly with homophilic datasets. Consequently, our analysis is focused on leveraging homophily within graphs. Yet, some graph datasets (Lim et al., 2021; Platonov et al., 2023) are heterophilic. In such datasets, while the label of each node does not tend to be similar to the label of its neighbors, the labels of neighboring nodes may still carry valuable information. Some of our analysis may

Table 5: On heterophilic graphs, distillation methods do not improve over standard MLP.

| Dataset | GCN | GLNN | MLP |
|---------|-----|------|-----|
| penn94 | 80.45 | 81.22 | 83.60 |
| pokec | 75.09 | 61.33 | 68.66 |

apply to heterophilic graphs. E.g., neighborhood histograms may carry information about the labels of the node, even when neighbors belong to different classes.

In Tab. 5, we evaluate our approach on the two heterophilic datasets used in the GLNN paper. We find that with no regularization a standard MLP outperforms reported distillation results. Furthermore, we found that regularization and label histograms did not improve upon the vanilla MLP. This supports our claim that current distillation methods are geared toward homophilic graphs, and that their performance is correlated with our explicit regularization.

**Dataset-specific variability.** While our explicit GNN-free components for optimizing MLPs show a high positive correlation to the performance of distillation methods, there are cases where they underperform. This variation suggests that different approaches might be influenced by dataset-specific characteristics. Further investigation into specific cases may allow future research to develop methods that enjoy the best of all worlds.

## 8 CONCLUSION

In this paper, we demystified the effectiveness of graph distillation methods. Our investigation centered on the hypothesis that GNNs enhance the optimization process of MLPs due to two factors: (1) GNNs serve as consistency regularizers, and (2) GNNs leverage unlabeled data effectively. To test this hypothesis, we devised a methodology that directly incorporates these properties without the utilization of GNNs, employing consistency loss and iterative pseudo-labeling. Moreover, we introduced an explicit method for incorporating structural features and demonstrated its comparable efficacy to learnable features.

## 9 SOCIETAL IMPACTS

This paper aims to advance the field of machine learning on graphs. It may improve social network analysis, recommendations, and the analysis of academic citation networks. However, similar to other works, risks include potential misuse for surveillance or privacy violations.

## 10 REPRODUCIBILITY

To ensure reproducibility, we ran each experiment 10 times and report the standard deviation. We describe the datasets we used in App. Sec. A.1, and provide implementation details in App. Sec. A.6. Additionally, we include our entire source code in the supplementary materials and will publish it on GitHub upon acceptance. The proof for Proposition 1, introduced in Sec. 4.3.1, is provided in App. Sec. A.2.

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

# A APPENDIX

## A.1 DATASETS

***Cora***, ***CiteSeer*** (Sen et al., 2008) and ***PubMed*** (Namata et al., 2012) are citation networks where each node represents a scientific paper, edges signify citations between papers, and labels denote the research field of each paper. In *Cora* and *CiteSeer* the feature vector of each node is a sparse bag-of-words derived from the text of the paper. *PubMed* is constructed from medical publications, the node features are represented by TF/IDF (Ramos et al., 2003) weighted word frequency.

***A-Computers*** and ***A-Photo*** (Shchur et al., 2018) are extracted from the Amazon co-purchase graph (McAuley et al., 2015). These datasets involve nodes representing electronic goods sold on Amazon web store. Edges indicate whether two products are frequently bought together. The node features are product reviews encoded using a bag-of-words representation. The labels assigned to the nodes correspond to product categories, with *A-Computers* encompassing categories such as Desktops, Laptops, Monitors, and so forth. *A-Photo* includes categories such as Cameras, Lenses etc.

***ogbn-arxiv*** and ***ogbn-products*** are from the Open Graph Benchmark (OGB) (Hu et al., 2020) and are larger datasets. The former constitutes a citation network of arXiv papers, while the latter is a co-purchasing network.

**Dataset split.** We follow the protocol used in previous studies for partitioning datasets into training, validation, and test sets. In the transductive setting, *Cora*, *CiteSeer*, *PubMed*, *A-Computers*, and *A-Photo* are partitioned by randomly sampling 20 instances per class for training, 30 instances per class for validation, and treating the remaining nodes as the test set. For the *ogbn-arxiv* dataset, the training is conducted on papers published until 2017, validation is performed on those published in 2018, and testing is carried out on papers published since 2019. In *ogbn-products*, nodes (representing products) are arranged based on their sales ranking. The top 8% of products are assigned

to the training set, the subsequent top 2% to the validation set, and the remaining to the test set. The partitioning scheme used by the OGB datasets is designed to perform an accurate simulation of real-life scenarios.

Table 6: Dataset Statistics.

| Dataset | # Nodes | # Edges | # Features | # Classes | Split (train / val / test) |
|---------|---------|---------|-----------|-----------|----------------------------|
| Cora | 2,485 | 5,069 | 1,433 | 7 | Random (140 / 210 / 2,135) |
| Citeseer | 2,110 | 3,668 | 3,703 | 6 | Random (120 / 180 / 1,810) |
| Pubmed | 19,717 | 44,324 | 500 | 3 | Random (60 / 90 / 19,567) |
| A-computer | 13,381 | 245,778 | 767 | 10 | Random (200 / 300 / 12,881) |
| A-photo | 7,487 | 119,043 | 745 | 8 | Random (160 / 240 / 7,087) |
| Arxiv | 169,343 | 1,166,243 | 128 | 40 | Public (53.7% / 17.6% / 28.7%) |
| Products | 2,449,029 | 61,859,140 | 100 | 47 | Public (8% / 1.6% / 90.4%) |

A.2  CONNECTION BETWEEN CONSISTENCY LOSS AND DISTILLATION

We provide a proof for the proposition presented in Sec. 4.3.1

**Proposition 1.** *Let $\Psi$ be a linear classifier with weights $W$. Define a corresponding GNN model, $\Phi$, as a 1-layer GCN model[3] that uses the same weights $W$. Then the linear model $\Psi$ that minimizes the loss $\mathcal{L}_{consist}$ is equivalent to the model that minimizes the GNN distillation loss, where the GNN model $\Phi$ acts as the teacher and the linear model $\Psi$ acts as the student.*

*Proof.* Denote the $c$'th row of $W$ is $w_c$. Let us fix a node $v$ with feature vector $\mathbf{x}$.

The student model's prediction for the label of $v$ is $\Psi(v) = W\mathbf{x}$, where the logit of class $c$ is its $c$'th entry, i.e., $w_c \cdot \mathbf{x}$. The GCN model's prediction is obtained by averaging the features of the neighbors of $v$ and then passing this average to a linear layer, i.e., $\Phi(v) = W\widetilde{\mathbf{x}}$, where $\widetilde{\mathbf{x}} = \frac{1}{|N(v)|} \sum_{v_j \in N(v)} \mathbf{x}_j$.

Thus,

$$CE(\Phi(v), \Psi(v)) = CE(W\widetilde{\mathbf{x}}, W\mathbf{x}) \tag{10}$$

$$= -\sum_{c=1}^{C} (w_c \cdot \widetilde{\mathbf{x}}) \log(w_c \cdot \mathbf{x}) \tag{11}$$

$$= -\sum_{c=1}^{C} \left( w_c \cdot \frac{1}{|N(v)|} \sum_{v_j \in N(v)} \mathbf{x}_j \right) \log(w_c \cdot \mathbf{x}) \tag{12}$$

$$= \frac{1}{|N(v)|} \sum_{v_j \in N(v)} \left[ -\sum_{c=1}^{C} (w_c \cdot \mathbf{x}_j) \log(w_c \cdot \mathbf{x}) \right] \tag{13}$$

$$= \frac{1}{|N(v)|} \sum_{v_j \in N(v)} CE(W\mathbf{x}_j, W\mathbf{x}) \tag{14}$$

By summing over all the nodes we get

$$\underbrace{\sum_{v_i \in \mathcal{V}} CE(\Phi(v_i), \Psi(v_i))}_{\mathcal{L}_{distill}(\Psi)} = \underbrace{\sum_{v_i \in \mathcal{V}} \left( \frac{1}{|N(v_i)|} \sum_{v_j \in N(v_i)} CE(\Psi(v_j), \Psi(v_i)) \right)}_{\mathcal{L}_{consist}(\Psi)} \tag{15}$$

$\square$

---

[3]This procedure is discussed by Yang et al. (2022)

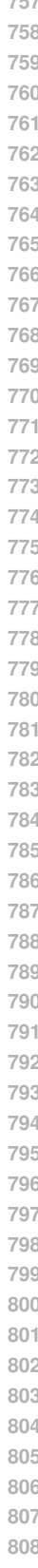

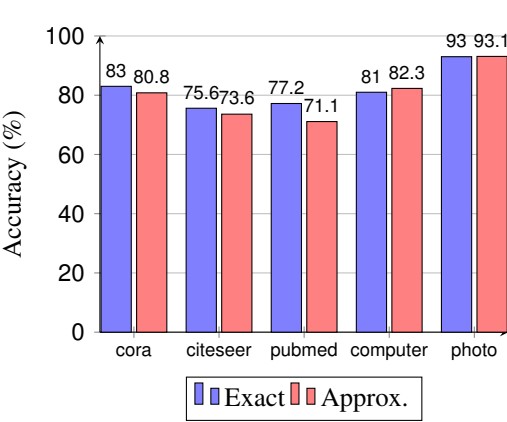

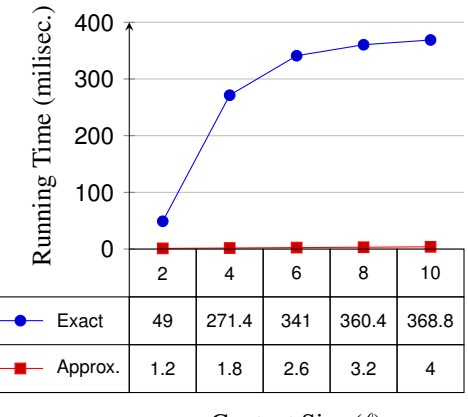

Figure 3: The model's accuracy is 1.66% higher on average when using the exact formula (Eq. 8) for histogram calculation compared to its approximation (Eq. 16).

Figure 4: Running times of the pre-processing procedure of calculating the histograms using the exact calculation (Eq. 8) and using its approximation (Eq. 16) as function as the context size.

### A.3 APPROXIMATING LABEL HISTOGRAMS IN LARGE GRAPHS

For larger datasets, such as *ogbn-products*, we use an efficient approximation for $\mathbf{h}'_i$ in Eq. 8 that is faster to compute using graph convolution operations. We calculate histogram for all nodes in the graph jointly by spreading the labels using convolution operations. Specifically, the matrix $H'$ whose rows represent un-normalized histograms for each node, is obtained by:

$$H' = \sum_{k=1}^{\ell} \left( \alpha \hat{A} \right)^k \tilde{Y} \tag{16}$$

Where the $i$-th row of $\tilde{Y}$ is defined by:

$$\tilde{\mathbf{y}}_i = \begin{cases} \mathbf{y}_i & v_i \in \mathcal{V}_{train} \\ \mathbf{0} & v_i \notin \mathcal{V}_{train} \end{cases}$$

We normalize $H'$ in the same manner presented in Eq. 9.

When using convolutions, this computation takes a running time of $\mathcal{O}(|E|)$. However, unlike the previous method of calculating histograms, the labels of some nodes in the training might leak into the label histogram feature. This can affect the generalization, as we do not have this information at test time. We observed that using small enough values for $\alpha$ eliminated the generalization gap due to this issue.

**Histogram approximation Ablation.** As described in Sec. 5, we augment the input features by concatenating histograms of labels derived from the local context of each node. Here, we analyze the trade-off between the time saved by the approximation and its potential impact on accuracy compared to exact computation. The results are presented in Fig. 3 and Fig. 4. The results show that while the exact histogram calculation results in a superior accuracy of 1.66% on average, approximating the histograms significantly reduces computation time.

### A.4 ADDITIONAL COMPARISONS

To expand our empirical study, in Tab. 7, we explore the following additional baselines, which use an MLP as a student model: FF-G2M (Wu et al., 2023a), KRD (Wu et al., 2023b), and CPF (Yang et al., 2021). This further evaluation supports our claim that the techniques we incorporate capture explicitly the regularization provided by GNN distillation.

Table 7: Comparisons to additional distillation methods.

| Dataset | SAGE | APPNP | GCN | GAT |
|---|---|---|---|---|
| cora | $80.52 \pm 1.77$ | $82.98 \pm 0.70$ | $81.82 \pm 1.26$ | $81.82 \pm 1.65$ |
| citeseer | $70.33 \pm 1.97$ | $72.70 \pm 1.53$ | $71.19 \pm 1.62$ | $71.47 \pm 1.75$ |
| pubmed | $75.39 \pm 2.09$ | $76.61 \pm 2.64$ | $76.73 \pm 2.64$ | $75.38 \pm 1.85$ |
| a-computer | $82.97 \pm 2.16$ | $81.15 \pm 2.57$ | $82.74 \pm 1.52$ | $83.12 \pm 1.54$ |
| a-photo | $90.90 \pm 0.84$ | $90.36 \pm 1.89$ | $91.18 \pm 0.84$ | $91.30 \pm 0.92$ |
| Mean | $80.02 \pm 1.83$ | $80.76 \pm 1.87$ | $80.73 \pm 1.58$ | $80.62 \pm 1.54$ |

| Dataset | GLNN-SAGE | GLNN-APPNP | GLNN-GCN | GLNN-GAT |
|---|---|---|---|---|
| cora | $80.54 \pm 1.35$ | $79.68 \pm 1.02$ | $80.53 \pm 1.57$ | $79.97 \pm 1.52$ |
| citeseer | $71.77 \pm 2.01$ | $72.96 \pm 1.64$ | $71.54 \pm 1.47$ | $71.96 \pm 1.71$ |
| pubmed | $75.42 \pm 2.31$ | $76.64 \pm 2.60$ | $78.01 \pm 2.75$ | $76.77 \pm 1.66$ |
| a-computer | $83.03 \pm 1.87$ | $80.23 \pm 2.40$ | $83.16 \pm 1.49$ | $83.19 \pm 1.38$ |
| a-photo | $92.11 \pm 1.08$ | $91.27 \pm 1.64$ | $92.92 \pm 0.67$ | $92.79 \pm 0.91$ |
| Mean | $80.57 \pm 1.78$ | $80.16 \pm 1.86$ | $81.23 \pm 1.59$ | $80.94 \pm 1.44$ |

| Dataset | FF-G2M | KRD | CPF | Ours |
|---|---|---|---|---|
| cora | $81.58 \pm 1.01$ | $81.11 \pm 1.42$ | $80.85 \pm 1.64$ | $82.92 \pm 1.15$ |
| citeseer | $73.25 \pm 1.10$ | $72.17 \pm 1.78$ | $70.67 \pm 2.11$ | $75.64 \pm 1.68$ |
| pubmed | $77.68 \pm 2.25$ | $77.30 \pm 1.81$ | $76.27 \pm 1.82$ | $77.22 \pm 2.49$ |
| a-computer | $81.15 \pm 2.34$ | $81.10 \pm 2.75$ | $80.92 \pm 2.08$ | $81.03 \pm 1.60$ |
| a-photo | $92.60 \pm 0.32$ | $92.42 \pm 0.48$ | $90.14 \pm 1.13$ | $93.06 \pm 1.56$ |
| Mean | $81.25 \pm 1.62$ | $80.82 \pm 1.80$ | $79.77 \pm 1.79$ | $81.97 \pm 1.75$ |

## A.5 ADDITIONAL ABLATIONS

We start by examining our classifier with only the standard cross-entropy classification loss. We see from App.Tab. 9 that the results of this naive classifier are far from being competitive. We show that each of the analyzed components has a significant impact on the performance. As can be seen in App.Tab. 9, pairs of these components already achieve strong results. Yet, the best performance is achieved when using all 3 components.

**Influence of node degree.** We extend our analysis of the relationship between node degree and model performance to scenarios involving noisy labels. For this, we evaluate the model's accuracy for nodes with varying edge degrees when 20% of the train and validation labels are replaced with random (noisy) labels. The results, presented in App. Fig. 6 indicate that the impact of noise is relatively consistent across nodes of varying degrees. Furthermore, this effect qualitatively resembles the patterns observed when labels are not noisy.

**Correlation between accuracy and node homophily.** App. Fig. 7 examines the correlation between confidence assigned by the model to the true label and the homophily rate of the node. This analysis reveals that the correlation observed for our MLP is more similar to that of GNN distillation than to a standard MLP, further supporting our hypothesis.

**Hyper-parameter sensitivity.** The consistency loss term is incorporated into the overall loss function with a weighting factor, denoted as $\gamma$. To assess the impact of this factor on model performance, we evaluate the model's accuracy across different values of $\gamma$. As shown in App. Fig. 8, the results indicate that this loss demonstrates robustness across a wide range of $\gamma$ values.

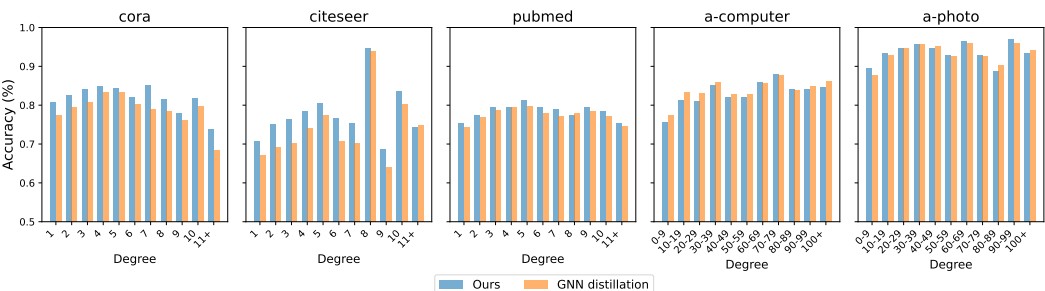

Figure 5: Accuracy vs. degree of nodes. The relation between the model performance and the node's degree exhibits a comparable pattern across both our MLP and GLNN (GNN distillation).

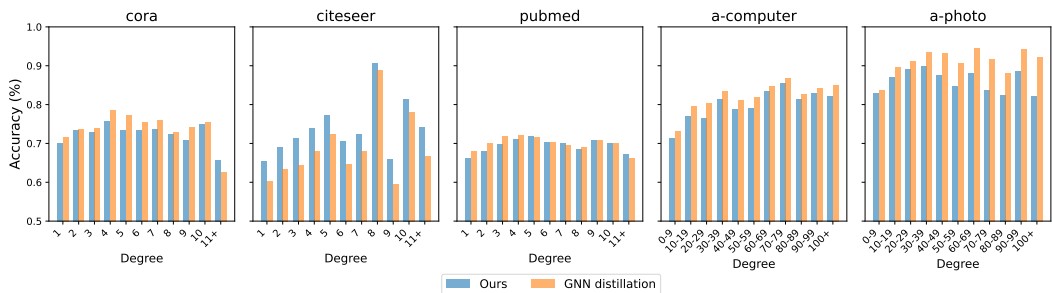

Figure 6: Accuracy vs. degree of nodes with noisy labels. We replace 20% of the train and validation labels with random labels. The relation between the model performance and the node's degree exhibits a comparable pattern across both our MLP and GLNN.

**Label histograms as classifiers.** To further analyze the importance of the histogram, we evaluate the accuracy of predictions based only on the most frequent label in the histogram, without any training. This is compared across different values of $\ell$, which represents the context size of the histogram. The results, presetned in App. Fig. 9, indicate that relying only on immediate neighbors ($\ell = 1$) is insufficient for accurate predictions. However, when the context size $\ell$ increases to 7, the accuracy is notably high for such a simplistic baseline approach.

## A.6 IMPLEMENTATION DETAILS

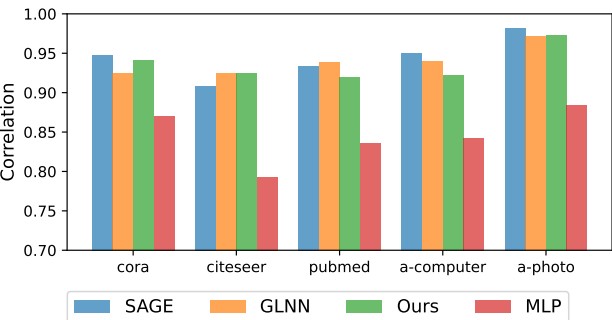

Figure 7: The correlation between confidence assigned by the model to the true label and the homophily rate of the node. Notably, the correlation exhibited by our MLP aligns more closely with that observed in GNN distillation processes than with that of a standard MLP.

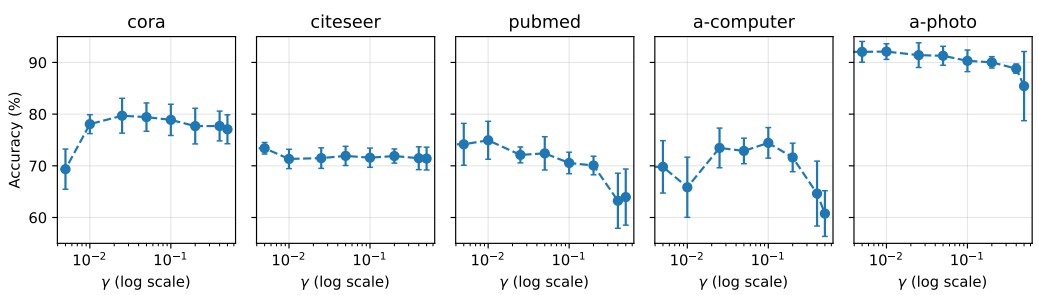

Figure 8: Accuracy vs. consistency regularization term ($\gamma$).

Table 8: Comparison between the accuracy on the test and train sets. MLPs have nearly 100% accuracy on the training set, while other models have lower training accuracy and higher testing accuracy.

| Dataset | MLP Train | MLP Test | SAGE Train | SAGE Test | GLNN Train | GLNN Test | Ours Train | Ours Test |
|---|---|---|---|---|---|---|---|---|
| cora | 100.0 | 46.3 | 99.4 | 81.0 | 96.1 | 80.3 | 99.7 | 83.0 |
| citeseer | 100.0 | 49.9 | 99.1 | 70.3 | 97.2 | 70.9 | 99.9 | 75.7 |
| pubmed | 100.0 | 64.6 | 99.8 | 75.5 | 98.7 | 76.4 | 99.8 | 77.2 |
| a-computer | 99.9 | 64.9 | 95.2 | 81.7 | 94.0 | 82.4 | 99.3 | 81.0 |
| a-photo | 100.0 | 73.2 | 97.9 | 90.9 | 96.7 | 92.5 | 99.6 | 93.1 |

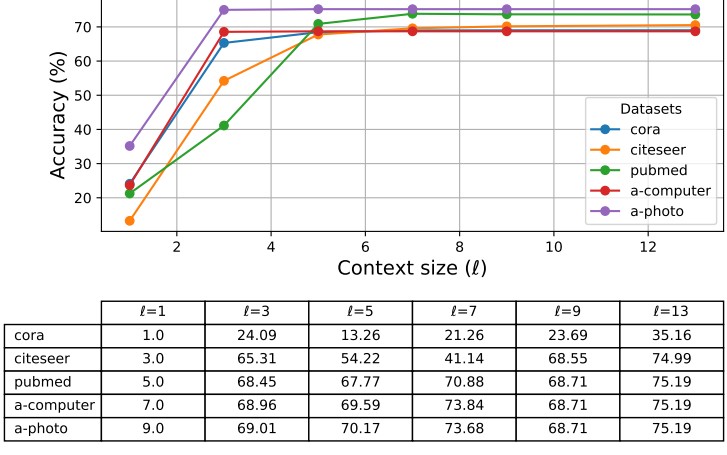

| | $\ell=1$ | $\ell=3$ | $\ell=5$ | $\ell=7$ | $\ell=9$ | $\ell=13$ |
|---|---|---|---|---|---|---|
| cora | 1.0 | 24.09 | 13.26 | 21.26 | 23.69 | 35.16 |
| citeseer | 3.0 | 65.31 | 54.22 | 41.14 | 68.55 | 74.99 |
| pubmed | 5.0 | 68.45 | 67.77 | 70.88 | 68.71 | 75.19 |
| a-computer | 7.0 | 68.96 | 69.59 | 73.84 | 68.71 | 75.19 |
| a-photo | 9.0 | 69.01 | 70.17 | 73.68 | 68.71 | 75.19 |

Figure 9: Accuracy of predictions based only on the most frequent label in the histogram, without any training. $\ell$ represents the context size of the histogram. Relying only on immediate neighbors ($\ell = 1$) is insufficient for accurate predictions. However, for values larger than 5 the accuracy is notably high for such a simplistic baseline.

Table 9: Ablation table

| Dataset | Base | Only Iterations | Only Histograms | Histograms and Consistency | Histograms and Iterations | Ours |
|---|---|---|---|---|---|---|
| cora | 56.88 | 77.72 | 78.86 | 81.75 | 81.79 | 82.92 |
| citeseer | 53.59 | 69.03 | 73.02 | 74.19 | 74.95 | 75.64 |
| pubmed | 64.03 | 65.89 | 75.77 | 76.57 | 77.04 | 77.22 |
| a-computer | 67.32 | 78.43 | 76.45 | 75.38 | 80.57 | 81.03 |
| a-photo | 77.75 | 88.42 | 86.59 | 90.21 | 91.31 | 93.06 |
| Mean | 63.91 | 75.90 | 78.14 | 79.62 | 81.13 | 81.97 |

---

**Algorithm 1** Pseudo-code

---

**Input:** Graph $\mathcal{G}$, Train-set $\mathcal{V}_{train}$, Iterations $T$, Confidence threshold $\tau$
Compute histograms according to Eq. 9 into $H$
$X \leftarrow \texttt{Concatenate}(X, H)$
Initialize $\Psi$, an MLP or linear model.
$\mathcal{V}_{train}^1 = \mathcal{V}_{train}$
**for** $t$ = 1 to $T$ **do**
    Train $\Psi$ on $\mathcal{V}_{train}^t$ and $\mathcal{G}$ using the loss from Eq. 5
    $\hat{Y} = \Psi(X)$
    $Y^* = \lambda \cdot \hat{Y} + (1 - \lambda) \cdot \hat{A}\hat{Y}$
    $\mathcal{V}_{train}^{t+1} \leftarrow \mathcal{V}_{train} \cup \{i| \max_{j=1,...,C}(Y_{ij}^*) > \tau\}$
**end for**

---

- We conducted each experiment using 10 different random seeds and reported the mean and standard deviation of the model accuracy on the test set.

- In the experiments of Sec 4, we use MLP with architecture of 2 layers and hidden dimensions of 128. In Sec. 5, the backbone consists of a single linear layer for all datasets, except for OGB datasets, where we utilized a two-layer MLP with hidden dimensions of 512 and 1024 for *ogbn-products* and *ogbn-arxiv*, respectively.

- Across all datasets, except for the OGB datasets, we employed 5 iterations, as outlined in Section 4.3.2. Within each iteration, the model underwent training for 200 epochs, and the optimal epoch was determined based on performance on the validation set. Notably, for the *ogbn-products* dataset, it was observed that a single epoch sufficed.

- On the larger datasets, *ogbn-products* and *ogbn-arxiv*, we observed that employing pseudo-labeling, as described in Sec. 4.3.2, is not necessary. This is primarily because they include large training splits, both in terms of the proportion of the entire graph, and in terms of the absolute number of labeled nodes. Consequently, in these cases, we report the performance only of the consistency loss and label-histogram features.

- In the iterative training described in Sec. 4.3.2, The weights of the model are initialized once, then at each iteration we continue the training for numerous epochs.

- The parameter defining the size of the local context utilized for computing the histogram, as explained in Section 5, was set to 10 hops (i.e., $\ell = 10$).

- We used Eq. 16 for calculation the approximation of the histograms in the larger datasets from OGB. While for all the other datasets we used the original formula (Eq. 8).

- The hyper-parameters were selected from the range presented in App.Tab. 10 via grid search.

**Python Libraries.** We use Deep Graph Library (DGL) (Wang et al., 2019a) for storing the graph datasets and performing graph operations on them. We also use PyTorch (Paszke et al., 2019) and scikit-learn (Pedregosa et al., 2011).

Table 10: Hyper-parameters Range

| Hyper-parameter | Search Range |
|---|---|
| Learning Rate | $[0.001, 0.005, 0.008, 0.01, 0.05, 0.1]$ |
| Weight Decay | $[0, 0.0001, 0.0005, 0.001, 0.005]$ |
| $\gamma$ (Eq. 5) | $[0.0, 0.005, 0.01, 0.025, 0.05, 0.1, 0.2, 0.4, 0.5]$ |
| $\tau$ (Eq. 6) | $[0, 0.2, 0.4, 0.5, 0.6, 0.8]$ |
| $\lambda$ (Eq. 7) | 0 to 1 in increments of 0.01 |
| $\alpha$ (Eq. 8) | $[0.025, 0.05, 0.1, 0.2, 0.4, 0.5, 0.6]$ |

## A.7 RUNNING TIMES

The label histograms described in Sec. 5 can be computed efficiently, requiring less than 5 milliseconds per node on a CPU, as shown in Fig. 4. For example, computing histograms for the large ogbn-products dataset (2.4M nodes) takes 1.26 minutes on an AMD 7713 processor. In contrast, the DeepWalk embeddings, proposed in previous work, requires over an hour on the same hardware.

In terms of training times, our approach directly trains the MLP, whereas distillation approaches involve first training a slower teacher GNN followed by the student MLP. This two-step process leads to longer overall training times. Tab. 11 summarizes the training times (averaged over 10 runs) on a single A10 GPU, demonstrating that our approach consistently achieves faster training across datasets.

Table 11: Training times in seconds.

| | Teacher SAGE | Student MLP | Distillation (teacher+student) | Ours |
|---|---|---|---|---|
| cora | 1.7 | 0.6 | 2.3 | 1.3 |
| citeseer | 1.5 | 0.5 | 2.0 | 0.9 |
| pubmed | 10.7 | 4.2 | 14.9 | 1.2 |
| a-computer | 15.2 | 4.5 | 19.7 | 3.1 |
| a-photo | 8.7 | 1.6 | 10.3 | 1.7 |

### A.7.1 INDUCTIVE SETTING

In the inductive setting we further split the unlabelled test set, denoted as $\mathcal{U} = \mathcal{V}/(\mathcal{V}_{train} \cup \mathcal{V}_{val})$, into two disjoint sets: (1) Unseen test nodes, a set of nodes exclusively available during inference time and not in training time, denoted by $\mathcal{U}_{unseen}$. (2) Observed test nodes, a set of unlabeled nodes with accessible features during training, denoted by $\mathcal{U}_{seen}$. Unlike the unseen test nodes, the observed test nodes participate in the consistency loss and may have pseudo-labels. In the inductive setting, we train the model on the graph induced by all the nodes in the set $\mathcal{V}_{train} \cup \mathcal{V}_{val} \cup \mathcal{U}_{seen}$. Only the nodes of $\mathcal{V}_{train}$ are used in the classification loss (Eq. 1). At training time, we discard edges that connect to nodes that are in $\mathcal{U}_{unseen}$. The test accuracy is computed on the combination of the sets $\mathcal{U}_{seen}$ and $\mathcal{U}_{unseen}$.

We evaluate the label-histogram descriptor described in Sec. 5 in the inductive settings. Similarly to the transductive case, using label-histogram achieve similar results to using distillation method that leverage learnable positional features, as shown in App.Tab. 12.

Table 12: Inductive setting: The test set is further partitioned into 80% test set that present during training (*seen*) and 20% unseen test (*unseen*). The formula used for computing a composite measure denoted as $prod$ is expressed as follows: $prod = 0.8 \cdot seen + 0.2 \cdot unseen$.

| Datasets | Eval | SAGE | MLP | GLNN | NOSMOG | Ours | $\Delta_{GLNN}$ | $\Delta_{NOSMOG}$ |
|---|---|---|---|---|---|---|---|---|
| Cora | *prod* | 79.53 | 59.18 | 78.28 | 81.02 | **81.11** | ↑ 2.83% | ↑ 0.09% |
| | *unseen* | 81.03 ± 1.71 | 59.44 ± 3.36 | 73.82 ± 1.93 | 81.36 ± 1.53 | 80.09 ± 2.29 | ↑ 6.27% | ↓ -1.27% |
| | *seen* | 79.16 ± 1.60 | 59.12 ± 1.49 | 79.39 ± 1.64 | 80.93 ± 1.65 | 81.37 ± 1.74 | ↑ 1.98% | ↑ 0.44% |
| Citeseer | *prod* | 68.06 | 58.49 | 69.27 | 70.60 | **72.94** | ↑ 3.67% | ↑ 2.34% |
| | *unseen* | 69.14 ± 2.99 | 59.31 ± 4.56 | 69.25 ± 2.25 | 70.30 ± 2.30 | 71.77 ± 3.37 | ↑ 2.52% | ↑ 1.47% |
| | *seen* | 67.79 ± 2.80 | 58.29 ± 1.94 | 69.28 ± 3.12 | 70.67 ± 2.25 | 73.23 ± 3.13 | ↑ 3.95% | ↑ 2.56% |
| Pubmed | *prod* | 74.77 | 68.39 | 74.71 | **75.82** | 74.51 | ↓ -0.2% | ↓ -1.31% |
| | *unseen* | 75.07 ± 2.89 | 68.28 ± 3.25 | 74.3 ± 2.61 | 75.87 ± 3.32 | 74.84 ± 3.39 | ↑ 0.54% | ↓ -1.03% |
| | *seen* | 74.70 ± 2.33 | 68.42 ± 3.06 | 74.81 ± 2.39 | 75.80 ± 3.06 | 74.43 ± 3.20 | ↓ -0.38% | ↓ -1.37% |
| A-computer | *prod* | 82.73 | 67.62 | 82.29 | **83.85** | 80.66 | ↓ -1.63% | ↓ -3.19% |
| | *unseen* | 82.83 ± 1.51 | 67.69 ± 2.20 | 80.92 ± 1.36 | 84.36 ± 1.57 | 80.19 ± 2.65 | ↓ -0.73% | ↓ -4.17% |
| | *seen* | 82.70 ± 1.34 | 67.60 ± 2.23 | 82.63 ± 1.4 | 83.72 ± 1.44 | 80.78 ± 2.29 | ↓ -1.85% | ↓ -2.94% |
| A-photo | *prod* | 90.45 | 77.29 | 92.38 | **92.47** | 92.11 | ↓ -0.27% | ↓ -0.36% |
| | *unseen* | 90.56 ± 1.47 | 77.44 ± 1.50 | 91.18 ± 0.81 | 92.61 ± 1.09 | 91.75 ± 1.36 | ↑ 0.57% | ↓ -0.86% |
| | *seen* | 90.42 ± 0.68 | 77.25 ± 1.90 | 92.68 ± 0.56 | 92.44 ± 0.51 | 92.20 ± 1.35 | ↓ -0.48% | ↓ -0.24% |
| Arxiv | *prod* | 70.69 | 55.35 | 65.09 | 70.90 | **71.32** | ↑ 6.23% | ↑ 0.42% |
| | *unseen* | 70.69 ± 0.58 | 55.29 ± 0.63 | 60.48 ± 0.46 | 70.09 ± 0.55 | 71.42 ± 0.34 | ↑ 10.94% | ↑ 1.33% |
| | *seen* | 70.69 ± 0.39 | 55.36 ± 0.34 | 71.46 ± 0.33 | 71.10 ± 0.34 | 71.29 ± 0.22 | ↓ -0.17% | ↑ 0.19% |
| Products | *prod* | 76.93 | 60.02 | 75.77 | 77.33 | **81.28** | ↑ 5.51% | ↑ 3.95% |
| | *unseen* | 77.23 ± 0.24 | 60.02 ± 0.09 | 75.16 ± 0.34 | 77.02 ± 0.19 | 81.69 ± 0.19 | ↑ 6.53% | ↑ 4.67% |
| | *seen* | 76.86 ± 0.27 | 60.02 ± 0.11 | 75.92 ± 0.61 | 77.41 ± 0.21 | 81.18 ± 0.18 | ↑ 5.26% | ↑ 3.77% |
| **Mean** | *prod* | 77.59 | 63.76 | 76.83 | 78.86 | **79.13** | ↑ 2.31% | ↑ 0.28% |

