# OpenReview forum: "Demystifying GNN Distillation by Replacing the GNN"
_ICLR.cc/2025/Conference — Submitted to ICLR 2025_

### Official Review · Reviewer_65su · 2024-10-17

**Soundness:** 2
**Presentation:** 3
**Contribution:** 2
**Rating:** 5
**Confidence:** 5

**Summary:**

This paper focuses on demystifying the role of Graph Neural Networks (GNNs) in graph distillation methods for graph node classification. It is found that GNNs primarily function as regularizers during distillation. Through theoretical and empirical investigations, the authors replace GNN distillation with explicit regularization strategies and propose label histogram as an alternative to positional embedding. The paper also validates its claims through comprehensive experiments on multiple datasets and comparisons with existing state-of-the-art methods.

**Strengths:**

1. The research perspective of the paper is interesting. It demonstrates that GNNs in distillation methods act as regularizers, which is supported by experiments comparing GNN distillation with an alternative approach of training MLPs with direct regularization.
2. The replacement of implicit GNN regularization with explicit terms can achieve comparable results to GNN distillation without training GNNs.
3. The writing is clear and good. The descriptions and explanations of individual methods are easy to follow.

**Weaknesses:**

1. The analysis is mainly focused on homophily graphs, and the performance on heterophily graphs is not well explored. A proper addition to this part of the results would strengthen my confidence on this paper. Another important issue is that I feel that “neighborhood smoothing” is an overly strong assumption, which makes the proposed method less applicable.
2. Proposition 1 and its proof seem to be based only on a linear setting. However, the actual MLP contains nonlinear activation, and I'm curious if the GNN distillation loss and consistency loss are still equivalent?
3. I have noticed that the datasets used in this paper are actually small, and it is very necessary to validate the effectiveness of the method on larger-scale ogb-arxiv and ogb-products datasets.
4. Lack of comparison with important baselines mentioned in related work, e.g. VQGraph and FF-G2M, etc.
5. The technical contribution of the article as a whole is limited, because the three proposed methods (1) label consistency; (2) iterative pseudo-labeling; and (3) label histograms share similar ideas with many past works. Combining them is indeed effective but an incremental contribution.

**Questions:**

Please address my concerns about weakness.

---

> ### Author Response · Authors · 2024-11-23
>
> Thank you for your detailed review. We appreciate that you found our paper interesting. We address each of your questions below:
>
>
> W1. ***“The analysis is mainly focused on homophily graphs”***
>
> As discussed in the limitations section, neither the GNN distillation methods nor the direct regularization technique we study, yield strong performance in scenarios characterized by heterophily. This study focuses on identifying the factors contributing to the success of distillation methods. Accordingly, we aim to use the same priors as distillation methods.
>
> ---
>
> W2. ***“Are GNN distillation loss and consistency loss equivalent?”***
>
> The GNN distillation loss and the consistency loss are (potentially) not mathematically equivalent in the non-linear activation. However, using a linearized network to analyze properties of DNNs is a common practice [1]. Furthermore, we would like to note the strong connection between our proof and our results. This theory motivated the connection between the consistency loss and the GNN; even though it is not necessarily a mathematical equivalence in the non-linear case. Empirical results support the connection in the non-linear case in Tab. 2.
>
>
> [1] Arora, Sanjeev, et al. "A convergence analysis of gradient descent for deep linear neural networks." arXiv preprint arXiv:1810.02281 (2018).
>
> ---
>
> W3. ***“it is necessary to validate the effectiveness on larger-scale datasets”***
>
> In Tab. 3 in the original manuscript, we compare our approach on much larger datasets: ogb-arxiv and ogb-products. We also include the results in a table below.
> |          | SAGE         | GLNN         | NOSMOG       | Ours         |
> |----------|--------------|--------------|--------------|--------------|
> | Arxiv    | 70.92 ± 0.17 | 72.15 ± 0.27 | 71.65 ± 0.29 | 71.35 ± 0.25 |
> | Products | 78.61 ± 0.49 | 77.65 ± 0.48 | 78.45 ± 0.38 | 81.71 ± 0.26 |
>
> ---
>
> W4. ***“Lack of comparison with important baselines”***
>
> We provided comparisons to additional methods in the Appendix Tab. 7 in the original manuscript.
> |            | SAGE         | FF-G2M       | KRD          | CPF          | Ours         |
> |------------|--------------|--------------|--------------|--------------|--------------|
> | Cora       | 80.52 ± 1.77 | 81.58 ± 1.01 | 81.11 ± 1.42 | 80.85 ± 1.64 |**82.92 ± 1.15** |
> | CiteSeer   | 70.33 ± 1.97 | 73.25 ± 1.1  | 72.17 ± 1.78 | 70.67 ± 2.11 | **75.64 ± 1.68** |
> | PubMed     | 75.39 ± 2.09 | **77.68 ± 2.25** | 77.30 ± 1.81 | 76.27 ± 1.82 | 77.22 ± 2.49 |
> | A-Computer | **82.97 ± 2.16** | 81.15 ± 2.34 | 81.10 ± 2.75 | 80.92 ± 2.08 | 81.03 ± 1.60 |
> | A-Photo    | 90.90 ± 0.84 | 92.60 ± 0.32 | 92.42 ± 0.48 | 90.14 ± 1.13 | **93.06 ± 1.56** |
> | Mean       | 80.02 ± 1.83 | 81.25 ± 1.62 | 80.82 ± 1.80 | 79.77 ± 1.79 | **81.97 ± 1.75** |
>
> ---
>
> W5. ***“Combining them is indeed effective but an incremental contribution.”***
>
> Although the individual components - consistency loss, iterative pseudo-labeling, and label histograms - may have similarities with existing methods, our work provides the novel insight that the success of GNN distillation methods hinges on regularization rather than expressivity. Through our analysis, we demonstrate that integrating these techniques into an MLP model produces results that closely align with those achieved through GNN distillation. This finding highlights that GNNs contribute to optimizing MLPs by inducing homophilic regularization.
>
> ---
>
> Thank you very much, once again, for your excellent comments. We respectfully ask that if your concerns are addressed, please consider updating your score. If not, please let us know what can be further improved; we are happy to continue the discussion any time until the end of the discussion period. Thank you!

---

> > ### Comment · Reviewer_65su · 2024-11-25
> >
> > I have read both authors' responses and other reviewers' comments. I strongly encourage the authors to move the comparison in the appendix (W4) into the main paper. Furthermore, I still have concerns about the incremental contribution of the paper, and the regularization perspective is not novel. Thus, I still keep my score.

---

### Official Review · Reviewer_Za6z · 2024-10-28

**Soundness:** 3
**Presentation:** 2
**Contribution:** 2
**Rating:** 6
**Confidence:** 4

**Summary:**

This paper discusses the distillation process from GNN to MLP. Rather than suggesting a new technique, the authors aim to demystify GNN distillation methods. They propose two hypotheses:(1) GNNs serve as consistency regularizers, and (2) GNNs leverage unlabeled data effectively. Their experimentals indicate that the distillation process of GNN2MLP can be replaced by other manually designed methods (without the need for GNN)

**Strengths:**

1. To understanding the process of GNN2MLP is quite important. The previous methods empirically confirmed the distillation process is  effective, but this work for the first time summarizes them into a unified hypothesis and verifies it through experiments.

2. The MLP trained by the author achieved performance comparable to GNN on different dataset sizes

**Weaknesses:**

1. All hypotheses have been experimentally proven, not theoretically. And the author only reported accuracy and lacked further analysis of the experimental results

2. During the iteration process, it seems that the nodes with pseudo-labels are constantly being replaced into the training set, and these nodes appear to come from outside the training set, which may lead to unfairness in the evaluation.

**Questions:**

1. During the iteration process, where does the node  with pseudo-labels come from? Will they be used for testing?


2. In Section 4.3.1, is the consistency loss highly correlated with the datasets? Using such a loss on the high homophily dataset used in the experiment seems to easily improve performance. Can you compare [homophily prediction encouraged by GNN during distillation] with [ homophily prediction encouraged by your suggested loss during MLP training]?


3. Can there be more analysis in the experiment, for example, on which specific nodes did your proposed method perform well? Is the performance improvement node brought by your method consistent with the performance improvement node brought by the GNN2MLP distillation process?

4. Line 343, "As our goal is to understand GNN distillation, we want to optimize the MLP model such that during inference we do not utilize neighboring nodes. Thus, we apply prediction smoothing only to the pseudo-labels within the iterative training algorithm." What is the causal relationship here?

Overall, I think the perspective of this paper is reasonable and interesting. But the manuscript is worth further revisions, and if the author can address my concerns, I am willing to increase my score.  :  )

---

> ### Author Response · Authors · 2024-11-23
>
> Thank you for your detailed review. We appreciate that you found our paper interesting and important. We address each of your questions below:
>
>
> W1. ***"... further analysis of the experimental results”***
>
> Q3. ***“Can there be more analysis in the experiment”***
>
> Following the reviewer’s comment, we extend in the revised version the empirical analysis with two additional experiments:
>
> 1. **Analysis of accuracy vs. node degree:** App. Fig. 5 illustrates the model's accuracy across nodes with varying edge degrees. The findings suggest that our optimization process for MLPs exhibits patterns closely aligned with those observed in GNN distillation. Additionally, we extend this analysis to scenarios involving noisy labels. For this, we evaluate the model’s accuracy for nodes with varying edge degrees when 20% of the train and validation labels are replaced with random (noisy) labels. The results, presented in App. Fig. 6, indicate that the impact of noise is relatively consistent across nodes of varying degrees. Furthermore, this effect qualitatively resembles the patterns observed when labels are not noisy.
>
> 2. **Correlation between accuracy and node homophily:**  App. Fig.  7 examines the correlation between confidence assigned by the model to the true label and the homophily rate of the node. This analysis reveals that the correlation observed for our MLP is more similar to that of GNN distillation than to a standard MLP, further supporting our hypothesis.
>
> ---
>
> W2. ***"these nodes appear to come from outside the training set”***
>
> Q1.***“where does the node with pseudo-labels come from?”***
>
> We would like to clarify that our evaluations are in line with other works, and do not use the test labels. Similarly to most of the GNN models and distillation methods, we leverage un-labeled nodes that are available during the training phase. These nodes, also known as observed nodes, are incorporated into the training process by assigning pseudo-labels to the unlabeled nodes based on our own predictions (and not the ground-truth labels). Namely, our node predictions for which the model generates highly confident predictions.
>
> ---
>
> Q2. ***“is the consistency loss correlated with the datasets?”***
>
> We analyze how incorporating homophily priors influences model performance across various datasets in Sec. 6. The effectiveness of the consistency loss demonstrates variability depending on the dataset (see Tab. 2). Yet, it is helpful on almost all of them. Analysis of the dataset variability with the consistency loss combined with pseudo-label smoothing is shown in Tab. 4.
>
>
> Q2. ***“Can you compare [homophily prediction encouraged by GNN during distillation] with [ homophily prediction encouraged by your suggested loss during MLP training]?”***
>
> We added to the revised version an analysis on the correlation between confidence assigned by the model to the true label and the homophily rate of the node. App. Fig.  7 shows that the correlation observed for our MLP is more similar to that of GNN distillation than to a standard MLP, confirming our hypothesis. We include the results in the table below.
> |            | SAGE  | GLNN  | Ours  | MLP   |
> |------------|-------|-------|-------|-------|
> | cora       | 0.947 | 0.925 | 0.941 | 0.870 |
> | citeseer   | 0.908 | 0.924 | 0.924 | 0.793 |
> | pubmed     | 0.933 | 0.938 | 0.920 | 0.836 |
> | a-computer | 0.950 | 0.940 | 0.922 | 0.842 |
> | a-photo    | 0.982 | 0.972 | 0.973 | 0.884 |
>
>
> ---
>
> Q4. ***“Line 343, "As our goal is to understand GNN distillation, we want to optimize the MLP model such that during inference we do not utilize neighboring nodes. Thus, we apply prediction smoothing only to the pseudo-labels within the iterative training algorithm." What is the causal relationship here?”***
>
> Thank you for asking. We see how this sentence might be unclear and would like to clarify. As GNN-distillation methods aim to avoid using the graph-structure during inference, we also want to avoid using it. Therefore, we perform label smoothing (that relies on the graph structure) only during training.
> We clarify this in the revised manuscript.
>
> ---
>
> Thank you very much, once again, for your excellent comments. We respectfully ask that if your concerns are addressed, please consider updating your score. If not, please let us know what can be further improved; we are happy to continue the discussion any time until the end of the discussion period. Thank you!

---

> > ### Comment · Reviewer_Za6z · 2024-11-24
> > **Thanks for your response.**
> >
> > Thanks for addressing the concerns. I will increase my score.

---

### Official Review · Reviewer_2z4s · 2024-11-04

**Soundness:** 2
**Presentation:** 3
**Contribution:** 2
**Rating:** 3
**Confidence:** 5

**Summary:**

This paper studies the GNN-to-MLP distillation problem. This paper introduces a framework including three techniques: smoothing-regularization loss, label-histogram as a structural embedding, and iterative pseudo label smoothing. Considering the motivations, the proposed techniques are reasonable and effective. These techniques are based on the observations of the performance gap between GNNs and MLPs with varying label rates. The authors also claim that GNN distillation loss could be equivalent to smoothing regularization loss under certain conditions. The experiments are conducted on several datasets and the results show that the proposed techniques are effective. The ablation study also shows the effectiveness of each technique.

**Strengths:**

1. The insight that distillation loss could be equivalent to smoothing regularization loss is interesting.

2. The writing is clear and straightforward. One can easily follow the paper.

3. Label histogram as a structural embedding seems much more effective and reasonable than the deepwalk embedding used in NOSMOG.

4. Notable improvements in performance are observed with the proposed techniques in some datasets.

**Weaknesses:**

1. (Soundness) Several claims are not well supported or not appropriate. For example, in lines 132-133, the authors claim that "MLPs overfit" with observations that MLPs perform poorly in few-labels setting. As far as I know, overfitting describes the performance gap between training and test sets, which is not the case here. The authors could provide the train-test performance gaps of GNNs and MLPs to support this claim. Another example is the claim in lines 139-140 that "distillation overcomes it by increasing the size of the training set with GNN pseudo-labels." This claim is not well supported, since for distillation, not only more pseudo-labels are introduced, but also the information inside these pseudo-labels is distinctive to what MLP itself could provide. The result of an additional ablative experiment in which GNN pseudo-labels are replaced with MLP's own pseudo-labels (in a self-training way) may help to validate this claim in a more convincing way.

2. (Novelty) Regularizing the MLP using graph structure is not a new idea. Some related works [1,2] are missing in this topic. Furthermore, the smoothing-regularization loss seems equivalent to the contrastive loss of [4] under the homophily assumption. The authors should make a distinction between these works and their work.

3. (Novelty) The iterative pseudo-labeling is also a common practice [5] in topics of GNN few-labels learning. Furthermore, this technique lacks sufficient intuition since the topic of this paper is distillation, not few-labels learning. The authors should provide more insights on why this technique is effective in distillation, especially in a normal setting where labels are sufficient.

4. (Experiment) As far as I know, the main focused benchmark in GNN-to-MLP distillation is the inductive setting. However, the authors put this important inductive setting in the appendix while the main experiments are conducted in the transductive setting. Furthermore, the proposed method is even surpassed by the weak baseline GLNN in some datasets in this setting, which makes it hard to believe the effectiveness of the proposed techniques.

5. (Experiment) Only GraphSAGE is used as the GNN model. The authors should provide more GNN teachers such as GCN, GAT, APPNP, etc. to make the results more convincing.

6. (Contribution) Most of the insights and techniques heavily rely on the assumption of homophily, which limits the contribution of the paper.






[1] Wu, T., Zhao, Z., Wang, J., Bai, X., Wang, L., Wong, N., & Yang, Y. (2023). Edge-free but structure-aware: Prototype-guided knowledge distillation from gnns to mlps. arXiv preprint arXiv:2303.13763.

[2] Zhang, H., Wang, S., Ioannidis, V. N., Adeshina, S., Zhang, J., Qin, X., ... & Yu, P. S. (2023). Orthoreg: Improving graph-regularized mlps via orthogonality regularization. arXiv preprint arXiv:2302.00109.

[3] Shin, Y. M., & Shin, W. Y. (2023). Propagate & Distill: Towards Effective Graph Learners Using Propagation-Embracing MLPs. arXiv preprint arXiv:2311.17781.

[4] Hu, Y., You, H., Wang, Z., Wang, Z., Zhou, E., & Gao, Y. (2021). Graph-mlp: Node classification without message passing in graph. arXiv preprint arXiv:2106.04051.

[5] Li, Q., Han, Z., & Wu, X. M. (2018, April). Deeper insights into graph convolutional networks for semi-supervised learning. In Proceedings of the AAAI conference on artificial intelligence (Vol. 32, No. 1).

**Questions:**

Most of the questions and suggestions are already mentioned in the weaknesses section. I would like to mention some minor points here.

1. The authors should use "GCN" explicitly instead of "GNN" in their claims, since they only use GCN architecture in the analysis and don't extend their analysis to general GNN models.

2. How does this method compute the label histogram of unobserved test nodes in the inductive setting? Will it access the graph during inference?

3. Inference time of this method should be discussed in the paper. Comparison of training costs would also be helpful.

4. I think Label histogram as a structural embedding is more important than the other two techniques, while little theoretical or empirical insight of it is discussed in the paper. More intuitions and studies on this technique would be helpful.

---

> ### Author Response · Authors · 2024-11-23
>
> Thank you for your very detailed review. We appreciate that you found our paper clear and interesting. We address each of your questions below:
>
> W1. ***“The authors could provide the train-test performance gaps of GNNs and MLPs to support this claim.”***
>
> We indeed claim that the few-labels setting significantly induces a performance gap between train and test sets. In the manuscript Fig. 1 shows that MLPs have similar accuracy to GNNs when the training set is bigger, while in smaller training sets MLPs are much worse. This indicates that MLPs overfit more than GNNs for small training sets.
> According to the reviewer's suggestion, to better support this claim, we add a comparison between the accuracy on the test and train sets in App. Tab 8. This table shows that MLPs have nearly 100% accuracy on the training set, while other models have lower training accuracy and higher testing accuracy. We also include the table below.
>
>
> ||MLP Train|MLP Test|SAGE Train|SAGE Test|GLNN Train|GLNN Test|Ours Train|Ours Test|
> |------------|:------:|:-----:|:-----:|:-----:|:-----:|:-----:|:-----:|:-----:|
> | cora       | 100.0 | 46.3 | 99.4 | 81.0 | 96.1 | 80.3 | 99.7 |   83.0  |
> | citeseer   | 100.0 | 49.9 | 99.1 | 70.3 | 97.2 | 70.9 | 99.9 | 75.7 |
> | pubmed     | 100.0 | 64.6 | 99.8 | 75.5 | 98.7 | 76.4 | 99.8 | 77.2 |
> | a-computer |  99.9 | 64.9 | 95.2 | 81.7 | 94.0 | 82.4 |  99.3 | 81.0 |
> | a-photo    | 100.0 | 73.2 | 97.9 | 90.9 | 96.7 | 92.5 | 99.6 | 93.1 |
>
>
>
> W1. ***“experiment in which GNN pseudo-labels are replaced with MLP's own pseudo-labels (in a self-training way) may help”***
>
> Our work indeed ablates the usage of pseudo-labels with experiments where we use the  MLP’s own pseudo-labels in a self-training approach, without any reliance on GNN-generated labels. As presented in Sec. 4.3.2, Table 2, this comparison demonstrates that using such MLP pseudo-labels improves the model's accuracy. Furthermore, additional supporting ablation studies are provided in App. Tab. 9 to reinforce this finding.
>
>
> ---
>
> W2. ***“Some related works are missing”***
>
> Thank you for bringing these works to our attention. We discuss them in the revised manuscript.
>
> ***“the smoothing-regularization loss seems equivalent to the contrastive loss of [4] …  The authors should make a distinction between these”***
>
> We thank the reviewer for this suggestion. Indeed there is a distinction, our loss only encourages similarity of neighboring nodes, and is not contrastive. Yet, we agree there is a relevant similarity and added an explanation on the distinction to the revised manuscript.
>
> ---
>
> W3. ***“topic of this paper is distillation, not few-labels learning”***
>
> ***“The authors should provide more insights on why this technique is effective in distillation, especially in a normal setting where labels are sufficient.”***
>
> The topic of our paper is indeed understanding GNN distillation. GNN distillation is studied mostly for the few-labels setting [1, 2].
>
> GNN distillation is mostly needed in cases where labels are scarce. In Fig. 1 we show that as the size of the training set increases, the performance gap between MLPs and GNN distillation diminishes. We show that part of the effectiveness of GNN distillation can be attributed to the utilization of unlabeled nodes. In our work, we directly emulate the key factors contributing to the success of GNN distillation on MLPs by using pseudo-labels. Specifically, we demonstrate that iterative pseudo-labeling achieves comparable performance to GNN distillation (see Tab. 2) when combined with a consistency loss that replicates the distillation loss - without requiring the GNN.
>
> ---
>
> W4. ***“the authors put this important inductive setting in the appendix“***
>
> Following previous works [1, 2], we first report the results on the transductive settings, and later on the inductive settings. Due to space limitations, we put this experiment in the appendix. We added a clearer reference to it from the main text.
>
> W4. ***“the proposed method is even surpassed by the weak baseline GLNN in some datasets”***
>
> The main goal of this paper is to analyze the key factors that make GNN distillation methods work. This includes, first, identifying the unique strengths of GNNs, namely that they act as homophily regularizers and that they benefit from unlabeled data. Second, we demonstrate that the performance of an MLP model that uses these properties is highly correlated with the performance of distillation. While we demonstrate practical techniques, we do not aim to propose a new method, but rather to provide a unique perspective and insights on training non-GNN graph classifiers.
>
> ---
>
> [1] Zhang, Shichang, et al. "Graph-less neural networks: Teaching old mlps new tricks via distillation.", 2021.
>
> [2] Tian, Yijun, et al. "Nosmog: Learning noise-robust and structure-aware mlps on graphs.", 2022.
>
> ---
>
> We continue our response in the following comment.

---

> > ### Author Response · Authors · 2024-11-23
> >
> > W5. ***“The authors should provide more GNN model”***
> >
> > Thank you for your suggestion. Similar to previous works [1,2], we compare to GraphSage as it is the teacher used in distillation baseline methods. Yet, following the reviewer request we are happy to report similar results for the GNNs models GCN, GAT and APPNP. We revised the manuscript to include these models in App. Tab. 7.
> >
> > | Dataset    | SAGE         | APPNP        | GCN          | GAT          | Ours         |
> > |------------|--------------|--------------|--------------|--------------|--------------|
> > | cora       | 80.52 ± 1.77 | 82.98 ± 0.70 | 81.82 ± 1.26 | 81.82 ± 1.65 | 82.92 ± 1.15 |
> > | citeseer   | 70.33 ± 1.97 | 72.70 ± 1.53 | 71.19 ± 1.62 | 71.47 ± 1.75 | 75.64 ± 1.68 |
> > | pubmed     | 75.39 ± 2.09 | 76.61 ± 2.64 | 76.73 ± 2.64 | 75.38 ± 1.85 | 77.22 ± 2.49 |
> > | a-computer | 82.97 ± 2.16 | 81.15 ± 2.57 | 82.74 ± 1.52 | 83.12 ± 1.54 | 81.03 ± 1.60 |
> > | a-photo    | 90.90 ± 0.84 | 90.36 ± 1.89 | 91.18 ± 0.84 | 91.30 ± 0.92 | 93.06 ± 1.56 |
> > | Mean       | 80.02 ± 1.83 | 80.76 ± 1.87 | 80.73 ± 1.58 | 80.62 ± 1.54 | 81.97 ± 1.75 |
> >
> > ---
> >
> > W6. ***“Most of the insights and techniques heavily rely on the assumption of homophily”***
> >
> > As discussed in the limitations section, neither the GNN distillation methods nor the direct regularization technique yield strong performance in scenarios characterized by heterophily. This study focuses on identifying the factors contributing to the success of distillation methods. Accordingly, we aim to leverage the same priors that are leveraged by distillation methods.
> >
> > ---
> >
> > Q1. ***”The authors should use "GCN" explicitly”***
> >
> > Thank you for your suggestion. The previous works in the field, which we aim to analyze, already coined the term “GNN-distillation” and we therefore refer to them as such to avoid confusion. We clarify this in the revised manuscript mentioning GCN explicitly.
> >
> > We would like to stress that our analysis does not rely on any specific GNN.
> >
> > ---
> >
> > Q2. ***“How does this method compute the label histogram of unobserved test nodes”***
> >
> > For unobserved nodes (i.e., in the inductive settings), we compute the label histograms according to the train graph (the training nodes and their edges). As this requires access to the graph during inference, we propose this as an alternative to computing positional features for these nodes, proposed by NOSMOG; which also requires this kind of access.
> >
> > ---
> >
> >
> > Q3. ***”Inference time of this method should be discussed in the paper. Comparison of training costs would also be helpful.”***
> >
> > The label histograms can be computed in less than 5 milliseconds per node (see Fig. 4) on a CPU. In contrast to the compared GNN distillation methods, the label histograms do not necessitate the learning of embeddings . The MLP classifier is the same for both ours and the compared methods, and takes less than 0.5 milliseconds to run. We added a discussion on running times in the App. Sec. A.7.
> >
> > In terms of training times, our method directly trains the MLP, whereas distillation approaches involve first training a slower teacher GNN followed by the student MLP. This two-step process leads to longer overall training times. We added a table with the training times in seconds (App. Tab. 11), demonstrating that our approach consistently achieves faster training across datasets. We also include the table below.
> > |            | Teacher SAGE | Student MLP | Distillation (teacher+student) | Ours  |
> > |------------|:------------:|:-----------:|:------------------------------:|:-----:|
> > | cora       |      1.7     |     0.6     |               2.3              |  1.3  |
> > | citeseer   |      1.5     |     0.5     |               2.0              |  0.9  |
> > | pubmed     |     10.7     |     4.2     |              14.9              |  1.2  |
> > | a-computer |     15.2     |     4.5     |              19.7              |  3.1  |
> > | a-photo    |      8.7     |     1.6     |              10.3              |  1.7  |
> >
> > ---
> >
> > We continue our response in the following comment.

---

> > > ### Author Response · Authors · 2024-11-23
> > >
> > > Q4. ***“Label histogram is more important …More intuitions and studies on this technique would be helpful“***
> > >
> > > Thank you for your comment.
> > > The rationale for including the label histogram as an additional feature is rooted in the assumption that the labels of a node’s neighbors can provide valuable clues about the node's own label. To incorporate a broader context, we calculate the histogram of a node using all nodes within a distance $\ell$ from it.
> > >
> > > To further analyze the importance of the histogram, we have included an additional figure (App. Fig 9)  illustrating the accuracy of predictions based only on the most frequent label in the histogram, without any training. This is compared across different values of  $\ell$, which represents the context size of the histogram. The results indicate that relying only on immediate neighbors ($\ell=1$) is insufficient for accurate predictions. However, when the context size $\ell$ increases to 7, the accuracy is notably high for such a simplistic baseline approach.
> > >
> > > We added this discussion to the revised manuscript.
> > >
> > > |            | $\ell=1$ | $\ell=3$   | $\ell=5$  | $\ell=7$ | $\ell=9$ | $\ell=13$  |
> > > |----------|:--------:|:-----:|:-----:|:-----:|:-----:|:-----:|
> > > | cora       |   24.1  | 65.3 | 68.5 | 69.0 | 69.0 | 69.0 |
> > > | citeseer   |   13.3  | 54.2 | 67.8 | 69.6 | 70.2 | 70.5 |
> > > | pubmed     |   21.3  | 41.1 | 70.9 | 73.8 | 73.7 | 73.6 |
> > > | a-computer |   23.7  | 68.6 | 68.7 | 68.7 | 68.7 | 68.7 |
> > > | a-photo    |   35.2  | 75.0 | 75.2 | 75.2 | 75.2 | 75.2 |
> > >
> > >
> > > The contribution of label histograms to the model's performance is validated in Tab. 3. Furthermore, in App. Tab. 9, we present results for a model that uses only label histograms as additional features, excluding the other two components (iterative training and consistency loss).
> > >
> > > ---
> > >
> > > Thank you very much, once again, for your detailed review. We respectfully ask that if you feel more positive about our paper, to please consider updating your score. If not, please let us know what can be further improved; we are happy to continue the discussion any time until the end of the discussion period. Thank you!

---

> > > > ### Comment · Reviewer_2z4s · 2024-11-24
> > > >
> > > > Thanks for your efforts on tackling my concerns. Some of the concerns are addressed, while the followings are not:
> > > >
> > > > W2: While both [4] and your method could let representations of connected nodes be similar, [4] can also let representations of non-connected nodes be seperate. Does this mean that your proposed method is a reduced version of [4]?
> > > >
> > > > W3: "GNN distillation is mostly needed in cases where labels are scarce." Is this true? I don't see any connection between GNN distillation and GNN learning in few labels learning. Moreover, MLP with simple self-training setting may also produce good result, as I suggested in Official Review, which is not ablated in revised version.
> > > >
> > > > W4: In inductive setting, the proposed method is surpassed by NOSMOG in many datasets, which makes it hard to believe the effectiveness of the proposed techniques.
> > > >
> > > > W4: The interesting insight of this paper should be further exploited to develop a effective method. Beating the state-of-art methods would provide more convicing evidence on that the authors have really exploited this finding.
> > > >
> > > > W5: I suggest providing more GNN model as teachers, not as baselines.
> > > >
> > > > W6: This issue still limits the scope of this paper. You can provide evidence on your claim in rebuttal "neither the GNN distillation methods nor the direct regularization technique yield strong performance in scenarios characterized by heterophily". Even if this is true, the design of the proposed methods which explicitly depends on homophily is not a good design, since it could not extend to general graphs.
> > > >
> > > > Q1: Just one analysis on GCN is not sufficient. Generalization to other GNN models is also needed to validate your claims.

---

### Official Review · Reviewer_J7kx · 2024-11-04

**Soundness:** 3
**Presentation:** 3
**Contribution:** 3
**Rating:** 6
**Confidence:** 4

**Summary:**

The authors first investigate " Why are distillation methods so successful? ".

To determine whether GNNs ability comes from increased expressivity or a inductive bias, they analyze their performance against MLPs with varying training set sizes. The authors in their work show that GNNs primarily act as regularizers in distillation, rather than increasing model expressivity. Further, they show that GNNs benefit from unlabeled data through the message passing mechanism.

The paper proposes  method for node classification that replaces GNNs with regularization techniques: consistency loss and iterative pseudo-labeling. This approach leverages unlabeled data and captures structural information using label histograms.

The authors perform empirical evaluation on diverse datasets. They show their method performs empirically better than existing methods. Further, ablation is conducted for diff components to understand its importance.

Overall a good paper.

**Strengths:**

1. Interesting observation: The paper demystifies GNN distillation by highlighting their role as regularizers.

2. The proposed approach of label histogram and pseudo labeling is easy/simple and effective

3. The authors do highlight limitations of their work. ( homophilic prior vs heterophily).

4. code is provided

5. Ablation studies are also provided.

**Weaknesses:**

1. Would recommend authors to add homophily ratio for each dataset so that one can understand how the results vary with diff homophily rates.

2. Why is only GraphSage taken as base GNN in Table3? Why GNNs like GAT are not considered?

3. What is the impact of gamma in eq.3? impact of weightage to itself and neighbor?

4. Does the degree of a node have some role to play? Since label histogram and consistency loss use neighborhood data. How is the performance on nodes of different degrees? Are there any studies? How much is the impact of degree/ availability of neighborhood data etc. on performance. Are there any studies on noisy neighbors?

**Questions:**

See weakness section.

---

> ### Author Response · Authors · 2024-11-23
>
> Thank you for your detailed review. We appreciate that you found our paper good and interesting. We address each of your questions below:
>
>
> W1. ***“add homophily ratio for each dataset”***
>
> Thank you for your suggestion. We added to Tab. 4 two homophily metrics: edge and node homophily rates. We note that datasets with stronger homophily benefit more from the homophilic priors. We also include these metrics in the table below.
>
> |               | Edge homophily | Node homophily |
> |---------------|----------------|----------------|
> | cora          | 84.3%          | 86.2%          |
> | citeseer      | 79.5%          | 80.3%          |
> | pubmed        | 83.8%          | 86.5%          |
> | a-computer    | 78.3%          | 81.7%          |
> | a-photo       | 83.2%          | 86.1%          |
> | ogbn-arxiv    | 67.8%          | 70.7%          |
> | ogbn-products | 80.8%          | 81.7%          |
>
>
> W2. ***“Why is only GraphSage taken as base GNN”***
>
> Similar to previous works [1, 2], we compare to GraphSage as it is the teacher used in distillation baselines methods. Yet, following the reviewer request we are happy to report similar results for the GNN models GCN, GAT and APPNP. We revised the manuscript to include these models in App. Tab. 7. We show the table below:
>
> | Dataset    | SAGE         | APPNP        | GCN          | GAT          | Ours         |
> |------------|--------------|--------------|--------------|--------------|--------------|
> | cora       | 80.52 ± 1.77 | 82.98 ± 0.70 | 81.82 ± 1.26 | 81.82 ± 1.65 | 82.92 ± 1.15 |
> | citeseer   | 70.33 ± 1.97 | 72.70 ± 1.53 | 71.19 ± 1.62 | 71.47 ± 1.75 | 75.64 ± 1.68 |
> | pubmed     | 75.39 ± 2.09 | 76.61 ± 2.64 | 76.73 ± 2.64 | 75.38 ± 1.85 | 77.22 ± 2.49 |
> | a-computer | 82.97 ± 2.16 | 81.15 ± 2.57 | 82.74 ± 1.52 | 83.12 ± 1.54 | 81.03 ± 1.60 |
> | a-photo    | 90.90 ± 0.84 | 90.36 ± 1.89 | 91.18 ± 0.84 | 91.30 ± 0.92 | 93.06 ± 1.56 |
> | Mean       | 80.02 ± 1.83 | 80.76 ± 1.87 | 80.73 ± 1.58 | 80.62 ± 1.54 | 81.97 ± 1.75 |
>
> [1] Zhang, Shichang, et al. "Graph-less neural networks: Teaching old mlps new tricks via distillation." arXiv preprint arXiv:2110.08727 (2021).
>
> [2] Tian, Yijun, et al. "Nosmog: Learning noise-robust and structure-aware mlps on graphs." arXiv preprint arXiv:2208.10010 (2022).
>
>
> W3. ***“What is the impact of gamma in eq.3”***
>
> In the distillation loss (Eq. 3), gamma determines the relative importance assigned to the teacher GNN predictions and the ground truth labels. In our consistency loss (Eq. 5), gamma controls the weighting of the consistency regularization term. We added to the revision App. Fig. 8, which depicts the accuracy of the model vs. the value of gamma. We find that our method is robust to a large range of gamma values.
>
>
>
> W4. ***“Does the degree of a node have some role to play?”***
>
> ***“Are there any studies on noisy neighbors?”***
>
> We incorporated an analysis examining the impact of node degree on the model accuracy. In App. Fig. 5 in the revised manuscript, we present the model's accuracy for nodes with varying edge degrees. The results indicate that our optimization process for MLPs closely aligns with the patterns observed in GNN distillation. In the *a-computer* dataset, the accuracy appears to exhibit a stronger positive correlation with node degree compared to other datasets. Yet, generally there is no strong connection between the performance and the node degree across datasets.
>
>
> Following the reviewer's suggestion, we did a similar experiment when replacing 20% of the train and validation labels with random (noisy) labels. We see that the effect is roughly similar for nodes of various degrees, and is qualitatively similar to the non-noisy labels.  Please see the results in the revised paper (App. Fig. 6).
>
> ---
>
> Thank you very much, once again, for your excellent comments. We respectfully ask that if you feel even more positive about our paper, to please consider updating your score. If not, please let us know what can be further improved; we are happy to continue the discussion any time until the end of the discussion period. Thank you!

---

> > ### Comment · Reviewer_J7kx · 2024-11-25
> > **Thanks**
> >
> > I thank the authors for new experiments.
> > I find the paper good. I maintain my score.
> >
> > The new expts show stability of the proposed method.
> >
> > Thanks.

---

### Author Response · Authors · 2024-11-23

We thank all of the reviewers for their valuable feedback. This paper's goal was to understand how GNN distillation methods enhance MLPs. It found two mechanisms that achieve this: (1) GNNs serve as homophily regularizers and (2) distillation effectively leverages unlabeled data. To validate our claims, we conducted extensive experimental evaluations.

We appreciate that the reviewers found our perspective to be both interesting (J7kx, 2z4s, 65su) and important (Za6z). Notably, our proposed MLP approach achieved performance comparable to GNN-based distillation methods, without necessitating GNN training (65su, Za6z). Additionally, the simplicity and effectiveness of the proposed label histograms were acknowledged (J7kx, 2z4s), alongside the clear writing (2z4s, 65su).

In response to the reviewers’ suggestions, we have incorporated the following updates and additional analyses in the revised manuscript (with all new additions highlighted in green text):
- Analysis examining the impact of node degree on the model accuracy in App. Fig. 5
- Analysis on performance under noisy labels settings in App. Fig. 6.
- Correlation between accuracy and node homophily in App. Fig. 7.
- Added homophily rates for various dataset to Tab. 4.
- Added comparisons to additional GNN teacher models in App. Tab. 7
- Sensitivity to the hyper-parameter $\gamma$ in App. Fig. 8.
- Provided further evidence that MLPs tend to overfit more than GNNs and distillation methods in App. Tab. 8.
 - Further analysis on label histograms in App. Fig. 9.

We would be very happy to keep the discussion going, addressing any points that remain unclear, or any new suggestions. Thanks again for your suggestions!

---

### Meta-Review · Area_Chair_ivkm · 2024-12-22

**Metareview:**

This paper investigates how MLP itself can alter GNN distillation by using an *explicit* regularization such as consistency regularization and pseudo-labeling. This viewpoint is examined through some theoretical analysis and numerical analysis.

The question here is interesting to reveal the mechanism of GNN distillation. On the other hand, that kind of view point is no longer novel. Indeed, several literature has already addressed similar notions.
(i) More detailed discussion about the novelty of this work should be provided in more details. As some reviewers pointed out, this paper is missing some important relevant work. Hence, this paper does not necessarily give so much new insight to the community.
(ii) The ablation study provided in this paper is not strong. More carefully designed experiments are required to see which factor is actually affecting the result.
(iii) Theoretical part is not technically strong (only linear model is considered). Only homophily setting is covered.

For these reasons, I think this paper does not meet the standards for acceptance.

**Additional Comments On Reviewer Discussion:**

The authors gave extensive rebuttals. However, the concerns described in the meta-review were not resolved. Hence, the reviewers did not update their scores to support the paper's acceptance.

---

### Decision · Program_Chairs · 2025-01-22

Reject